# Profiling of insulin-resistant kidney models and human biopsies reveals common and cell-type-specific mechanisms underpinning Diabetic Kidney Disease

Abigail C. Lay [1,2], Van Du T. Tran [3], Viji Nair[4], Virginie Betin[1], Jennifer A. Hurcombe[1], Alexandra F. Barrington[1], Robert JP Pope[1], Frédéric Burdet [3], Florence Mehl [3], Dmytro Kryvokhyzha [5], Abrar Ahmad [5], Matthew C. Sinton [2], Philip Lewis[6], Marieangela C. Wilson [6], Rajasree Menon [4,7], Edgar Otto [4], Kate J. Heesom[6], Mark Ibberson [3], Helen C. Looker[8], Robert G. Nelson[8], Wenjun Ju [4,7], Matthias Kretzler[4,7], Simon C. Satchell [1], Maria F. Gomez [5], Richard J. M. Coward [1] ✉ & BEAt-DKD consortium*

Diabetic kidney disease (DKD) is the leading cause of end stage kidney failure worldwide, of which cellular insulin resistance is a major driver. Here, we study key human kidney cell types implicated in DKD (podocytes, glomerular endothelial, mesangial and proximal tubular cells) in insulin sensitive and resistant conditions, and perform simultaneous transcriptomics and proteomics for integrated analysis. Our data is further compared with bulk- and single-cell transcriptomic kidney biopsy data from early- and advanced-stage DKD patient cohorts. We identify several consistent changes (individual genes, proteins, and molecular pathways) occurring across all insulin-resistant kidney cell types, together with cell-line-specific changes occurring in response to insulin resistance, which are replicated in DKD biopsies. This study provides a rich data resource to direct future studies in elucidating underlying kidney signalling pathways and potential therapeutic targets in DKD.

Diabetic kidney disease (DKD) is the leading cause of end-stage kidney failure worldwide, occurring in up to 50% of individuals with diabetes[1]. Furthermore, the excess cardiovascular and all-cause mortality observed in individuals with diabetes is almost exclusively restricted to those with DKD[2,3]. Despite its prevalence, the molecular mechanisms underlying DKD development remain incompletely understood. Both glomerular and proximal tubular damage are hallmarks of DKD development and progression[4], thus, understanding the cellular and molecular changes occurring in both the glomerulus and proximal tubule is key to understanding the mechanisms underlying DKD, identifying therapeutic targets and biomarker candidates.

[1]Bristol Renal, Bristol Medical School, University of Bristol, Bristol, UK. [2]Division of Cardiovascular Sciences, Faculty of Biology, Medicine and Health, University of Manchester, Manchester, UK. [3]Vital-IT group, SIB Swiss Institute of Bioinformatics, Lausanne, Switzerland. [4]Division of Nephrology, Department of Internal Medicine, University of Michigan, Ann Arbor, MI, USA. [5]Department of Clinical Sciences, Lund University Diabetes Centre, Lund University, Malmö, Sweden. [6]Proteomics Facility, University of Bristol, Bristol, UK. [7]Department of Computational Medicine and Bioinformatics, University of Michigan, Ann Arbor, MI, USA. [8]Chronic Kidney Disease Section, National Institute of Diabetes and Digestive and Kidney Diseases, National Institute of Health, Phoenix, AZ, USA. *A list of authors and their affiliations appears at the end of the paper. ✉e-mail: Richard.Coward@bristol.ac.uk

Insulin resistance is one of the strongest metabolic features of DKD, in both type 1 and type 2 diabetes[5–8]. The re-classification of diabetic patients into subclusters (based on six variables, including systemic insulin resistance) has emphasised this relationship, highlighting that cellular insulin-resistant individuals at initial presentation have the highest risk of developing DKD[5] and indicating that kidney disease occurs secondary to insulin resistance[9].

Insulin can signal to multiple kidney cell types, including glomerular cells[10–12] (podocytes (Pods), mesangial cells (MCs), glomerular endothelial cells (GECs)) and proximal tubular cells[13–15] (PTCs). Consequently, any disruption to insulin signalling in these cells could potentially have important implications for kidney function in the early and later stages of DKD. Several studies have shown that a loss of intrinsic insulin signalling responses in kidney cells occurs in diabetes[11,16–18]. Furthermore, reduced cellular insulin signalling contributes to kidney injury[10,19] and can disrupt whole-body glucose homoeostasis[13].

We, therefore, aimed to further explore the changes occurring in kidney cells in response to a diabetic, insulin-resistant environment to highlight key pathways and processes linked to DKD pathogenesis. We performed comprehensive transcriptome and proteome analysis on human insulin-sensitive and insulin-resistant Pods, GECs, MCs, and PTCs to determine the molecular changes occurring in these conditions and aid our understanding of disease mechanisms. The use of in vitro cell lines as our model system allowed us to isolate RNA and protein simultaneously from the same population of cells, allowing the direct comparison of transcriptomic and proteomic alterations and performing functional studies to validate the results of gene set enrichment and pathway analysis.

We further explored our findings in kidney biopsies from human DKD cohorts, highlighting key gene expression and pathway changes for follow-up and demonstrating the utility of these cell models for future mechanistic studies of kidney cell dysfunction in DKD.

## Results

### Exposure to an in vitro diabetic environment causes insulin resistance in human kidney cells

An overview of our experimental design is presented in Fig. 1a. Previously characterised conditionally immortalised human kidney Pods[20], GECs[21], and MCs[22], together with PTCs[23], were used to model insulin sensitivity and insulin resistance. Consistent insulin sensitivity was achieved with stable insulin receptor (IR) over-expression, as prolonged culture of kidney cells in vitro can promote IR degradation[17]. Insulin resistance was induced by exposing cell lines to a 'diabetic milieu' ('DM') consisting of TNFα, IL-6, high glucose, and high insulin as previously described[17,24].

Initial characterisation of cellular insulin sensitivity demonstrated efficient expression of the IR in transduced cells and increased phosphorylation of IRβ (Tyr1150/1151) and Akt (S473) following insulin-stimulation of IR-expressing cells, which was lost in insulin-resistant conditions (Fig. 1b, Supplementary Fig. 1), as previously shown in Pods[17]. Interestingly, while IR protein levels were reduced in "diabetic" Pods[17], MCs and PTCs, this did not occur in GECs (Fig. 1b). Insulin-stimulated glucose uptake was also observed in IR-transduced GECs, Pods[17] and PTCs and lost following exposure to the diabetic, insulin resistant environment (Fig. 1c–f). Insulin stimulation did not induce an increase in cellular glucose uptake in MCs (Fig. 1e) and IR transduction alone had no significant effect on basal glucose uptake in our cell models (Supplementary Fig. 1m–p).

### Overview of the transcriptome and proteome changes occurring in insulin-resistant human kidney cells

To further explore the changes occurring in insulin-resistant kidney cells, the cellular proteome and transcriptome were studied simultaneously in each of the four kidney cell types, using tandem mass tagged (TMT)-based mass spectrometry and RNA sequencing. 6227 proteins and 18,359 transcripts were detected across all the cell types and conditions studied in the five independent experimental repeats. Principal component analysis (PCA) and heatmaps of the sample-to-sample distances performed on both the transcriptome and proteome (Fig. 2a, Supplementary Fig. 2a–c) demonstrated a grouping of samples primarily by cell type. The variation between insulin-sensitive and insulin-resistant cells was evident in PCAs performed within the individual cell populations (Supplementary Fig. 2d, e). *NPHS2, PECAM, EBF1* and *RGN* were examples of cell-specific genes that were solely detected in Pods, GEC, MCs and PTCs, respectively (Supplementary Fig. 3).

To identify the proteins and transcripts significantly regulated in an insulin-resistant environment, we initially performed a differential expression (DE) analysis, comparing insulin-resistant to insulin-sensitive cells. This indicated that, with the exception of GEC, there were more significantly regulated (false discovery rate [FDR]-adjusted $p < 0.05$) molecules in the IR-expressing cells exposed to an insulin-resistant environment vs wild-type cell lines (i.e., our highly insulin-sensitive vs insulin-resistant comparison) (Fig. 2b–e, Supplementary Fig. 4).

IR-expressing Pods and PTCs displayed the highest number of significantly regulated transcripts (1879 in Pods/1544 in PTCs) and proteins (291 in Pods/122 in PTCs) in response to insulin resistance (Fig. 2b–e). 79 DE transcripts/22 DE proteins and 579 DE transcripts/45 DE proteins were detected in insulin-resistant GEC and MCs, respectively. Examples of genes and proteins highly regulated in response to insulin resistance are labelled in Fig. 2c, d.

**Integrated analysis of the proteome and transcriptome highlights genes and proteins that are consistently regulated across insulin-resistant kidney cells and in human DKD.** To identify gene signatures that were consistently regulated across all insulin-resistant cells studied, we next performed integrated analyses of the proteome and transcriptome. We first examined the correlation between changes at the transcript and protein level for all protein-coding genes, where we had information on both transcript and protein abundance. Overall, there was a good correlation between regulated proteins and transcripts (Log2FC values) and a stronger correlation in the more insulin-sensitive IR-expressing cells ($r = 0.438–0.606$, Fig. 3a, Supplementary Fig. 5), which were subsequently focused on. This also allowed us to highlight examples of genes that were consistently regulated across all four cell types, at both the transcript and protein level, in response to insulin resistance (e.g., *SERPINB4*, Fig. 3a).

To capture additional changes in transcripts and proteins that were driven by insulin resistance, we further used a multivariate model; consensus orthogonal projections to latent structures discriminant analysis (OPLS-DA) (Supplementary Fig. 6), integrating both transcriptomic and proteomic data. The "Top 40" transcripts and proteins with evidence of consistent regulation across all insulin-resistant cell types (with variable importance in projection [VIP] > 1 and FDR < 0.1 in at least 3 comparisons) were selected for further evaluation (Fig. 3b). Additional details of these genes (including known functions) can be found in Supplementary Data 1. To investigate whether there were likely to be any shared transcription factors that could be driving these conserved gene expression changes, an over-representation analysis against the TRANSFAC database of eukaryotic transcription factors was performed. This suggested that these 40 commonly regulated genes were enriched for SRF, UBP1, PITX2 and MYB transcription factor binding sites (Supplementary Data 2).

**Evaluation of prioritised genes in biopsies from human DKD.** We next used transcriptomics data from human kidney biopsies to explore the regulation of these genes in early- or advanced-stage DKD and found that six of our "insulin-resistance-associated transcripts" were consistently up-regulated in both the glomerular and tubular regions

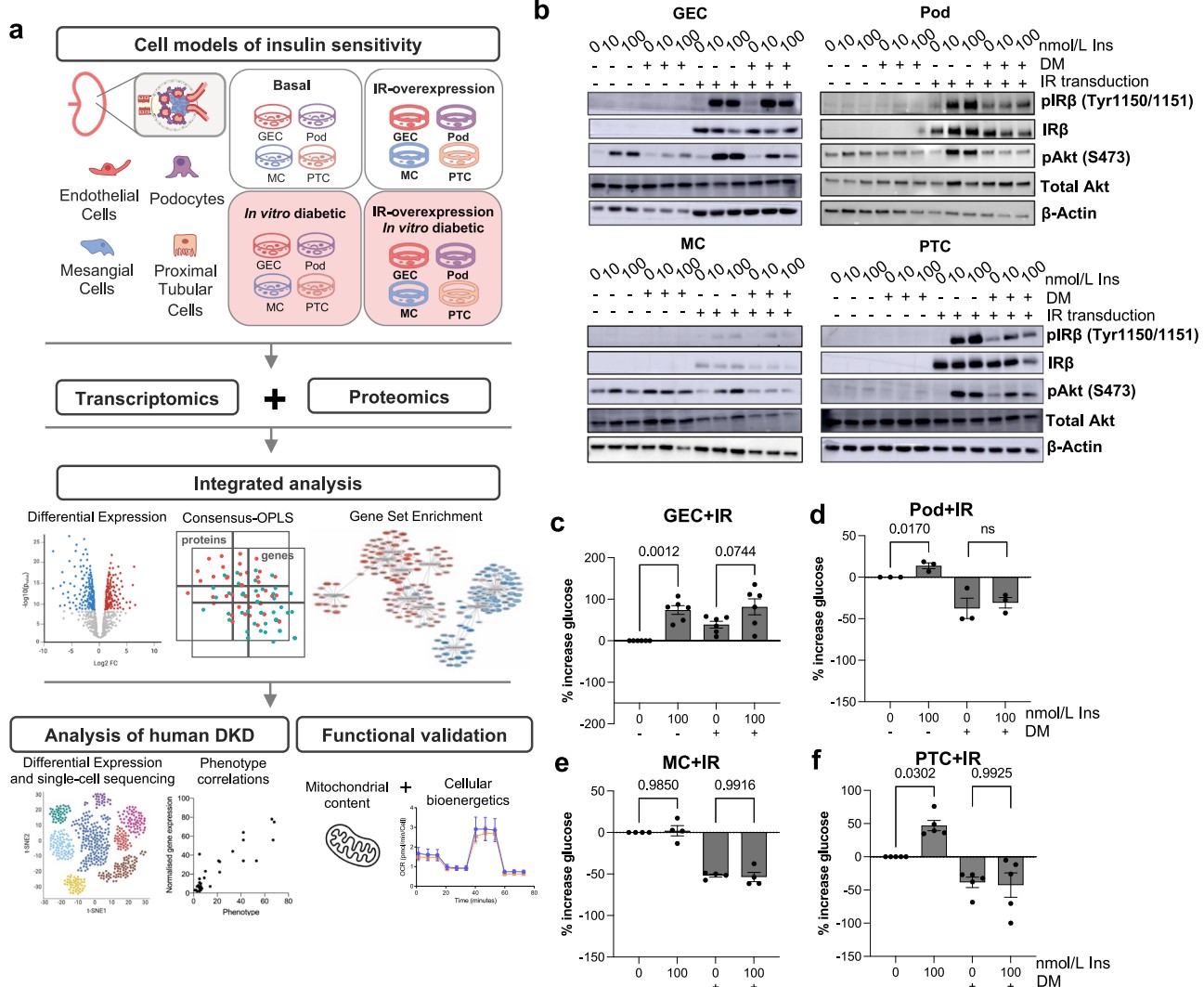

**Fig. 1 | Overview of experimental pipeline and characterisation of cell models.** **a** Schematic representation of the experimental pipeline. Conditionally immortalised human glomerular endothelial cells (GECs), podocytes (Pods), mesangial cells (MCs) and proximal tubular cells (PTCs) were studied in vitro in a basal and insulin-resistant environment (consisting of 1 ng/ml TNFα, 1 ng/ml IL-6, 25 mM glucose and 100 nmol/l insulin). Insulin-sensitive cell lines were established via stable over-expression of the human insulin receptor (IR). The cellular transcriptome and proteome were studied simultaneously using RNA sequencing and tandem-mass-tagged mass spectrometry ($n = 5$ biological repeats per cell line and condition) and integrated transcriptome and proteome data were analysed using univariate and multivariate statistical models and gene set enrichment analysis (GSEA). Further targeted analysis and validation were performed using single-cell and bulk transcriptomics data from human DKD biopsies. Figure partly created in BioRender. Lay, A. (2022) BioRender.com/x18l854 and BioRender. Lay, A. (2024) BioRender.com/m22n059. **b** Western blotting of total protein lysates demonstrated suppression of insulin-stimulated (15-min, 10 or 100 nmol/L) IR and Akt phosphorylation in all cell lines exposed to diabetic, insulin resistant, milieu ('DM'). GECs displayed no evidence of IR downregulation (representative of $n = 4$ biological replicates). **c–f** Percentage increase in cellular uptake of [3H]2-deoxy-ᴅ-glucose in insulin-stimulated (15-min, 100 nmol/L) GECs ($n = 6$), Pods ($n = 3$), MCs ($n = 4$) and PTCs ($n = 5$) [all biological repeats] vs. unstimulated cells, with and without exposure to a diabetic, insulin resistant, milieu ('DM'), unpaired two-tailed t-test, data are presented as mean values ± SEM.

of early and/or advanced human DKD (Fig. 3c); *C3* (encoding complement component 3), *CXCL1* (encoding Chemokine (C-X-C motif) ligand 1), *CTSS* (encoding Cathepsin S), *NRBF2* (encoding Nuclear Receptor-binding factor 2), *PFKFB3* (encoding 6-phosphfructo-2-kinase/fructose-2, 6-biphosphatase 3) and *TFPI2* (encoding tissue factor pathway inhibitor 2). Using longitudinal clinical information available from the early-stage type 2 diabetes cohort[25], we also observed significant correlations between the expression of some of these genes with kidney-associated phenotypes: *C3* (increased expression associated with faster rate of GFR decline i.e., 'slope'), *CTSS* (higher glomerular expression associated with higher albuminuria at time of biopsy and faster GFR decline) and *NRBF2* (higher expression associated with higher measured GFR at time of biopsy in individuals with

type-2 diabetes and early-stage DKD and subsequent faster rate of GFR decline) (Fig. 3d). Of note, *NRBF2* expression was also negatively correlated with GFR in late-stage DKD (Supplementary Fig 7), which may suggest that the modest positive correlations observed in early-stage type-2 DKD reflect hyperfiltration in those individuals (Supplementary Data 3; mean GFR was 145 ml/min in this group).

**NRBF2 loss is detrimental in all kidney cell types.** Given that *NRBF2* has no prior links to kidney function or insulin resistance, we next investigated the role of *NRBF2* in kidney cell lines. In all cell types studied (Pod, GECs, MCs and PTCs), *NRBF2* knock-down (Supplementary Fig. 8a) resulted in dramatic morphological differences in each cell type, including apparent vacuolisation (Fig. 4a) and a significant loss of

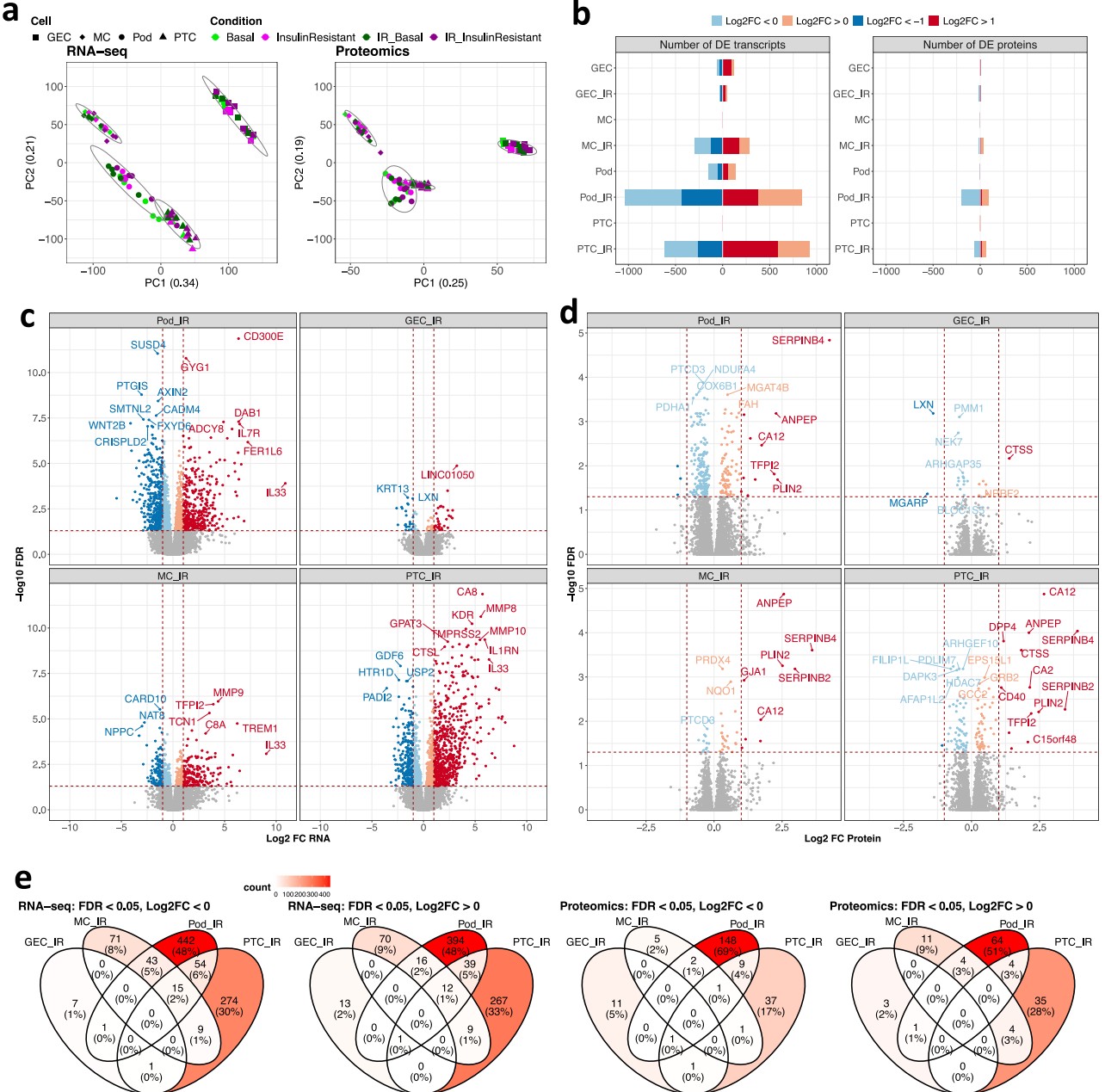

**Fig. 2 | Transcriptome and proteome changes in insulin-resistant kidney cell lines.** RNA and protein were simultaneously isolated from podocytes (Pods), glomerular endothelial cells (GECs), mesangial cells (MCs) and proximal tubular cells (PTCs) under basal and insulin-resistant conditions, with and without additional IR-transduction (*n* = 5 biological repeats/condition). **a** Principal component analysis (PCA) of >18,000 transcripts identified by RNAseq and >6000 proteins, detected across all cell lines and conditions studied, demonstrating the primary clustering of samples by cell type. **b** The number of differentially expressed (DE) transcripts and DE proteins (FDR < 0.05) within each cell type (both stable insulin receptor-expressing and non-insulin receptor transfected cells) in diabetic (insulin-resistant) vs. basal (insulin-sensitive) conditions. **c** Transcripts and **d** proteins differentially expressed IR-transduced cell lines in diabetic (insulin-resistant) vs. basal (insulin-sensitive) conditions, with examples of significantly regulated molecules highlighted (FDR < 0.05); differential expression analyses of full transcriptomics and proteomics datasets are available in the 'Source data' file. **e** Venn diagrams demonstrating the overlap of DE (FDR < 0.05, Log2FC < 0 or Log2FC > 0) transcripts or proteins between individual cell types, (analysis on IR-expressing cells alone).

GECs, MCs and PTCs within 96 h (Fig. 4b). In podocytes *NRBF2* knockdown resulted in fewer but larger podocytes (opposed to podocyte loss) (Fig. 4a, c). In contrast, *NRBF2* overexpressing podocytes (Supplementary Fig. 8b) were protected against actin cytoskeletal changes following exposure to an insulin-resistant environment (Fig. 4d, e). Collectively, these results indicate that altered *NRBF2* expression in Pods, GECs, MCs and PTCs has important functional consequences, which differ between kidney cell types.

## Enhanced inflammatory-response, ER-stress and glycoprotein metabolism are dysregulated pathways in all insulin-resistant kidney cell types

To provide further biological context to the changes occurring in insulin-resistant kidney cells, we performed a comprehensive gene set enrichment analysis (GSEA)[26], integrating data from both the transcriptome and the proteome. GO terms that were significantly enriched in at least one cell type at both RNA and protein level were

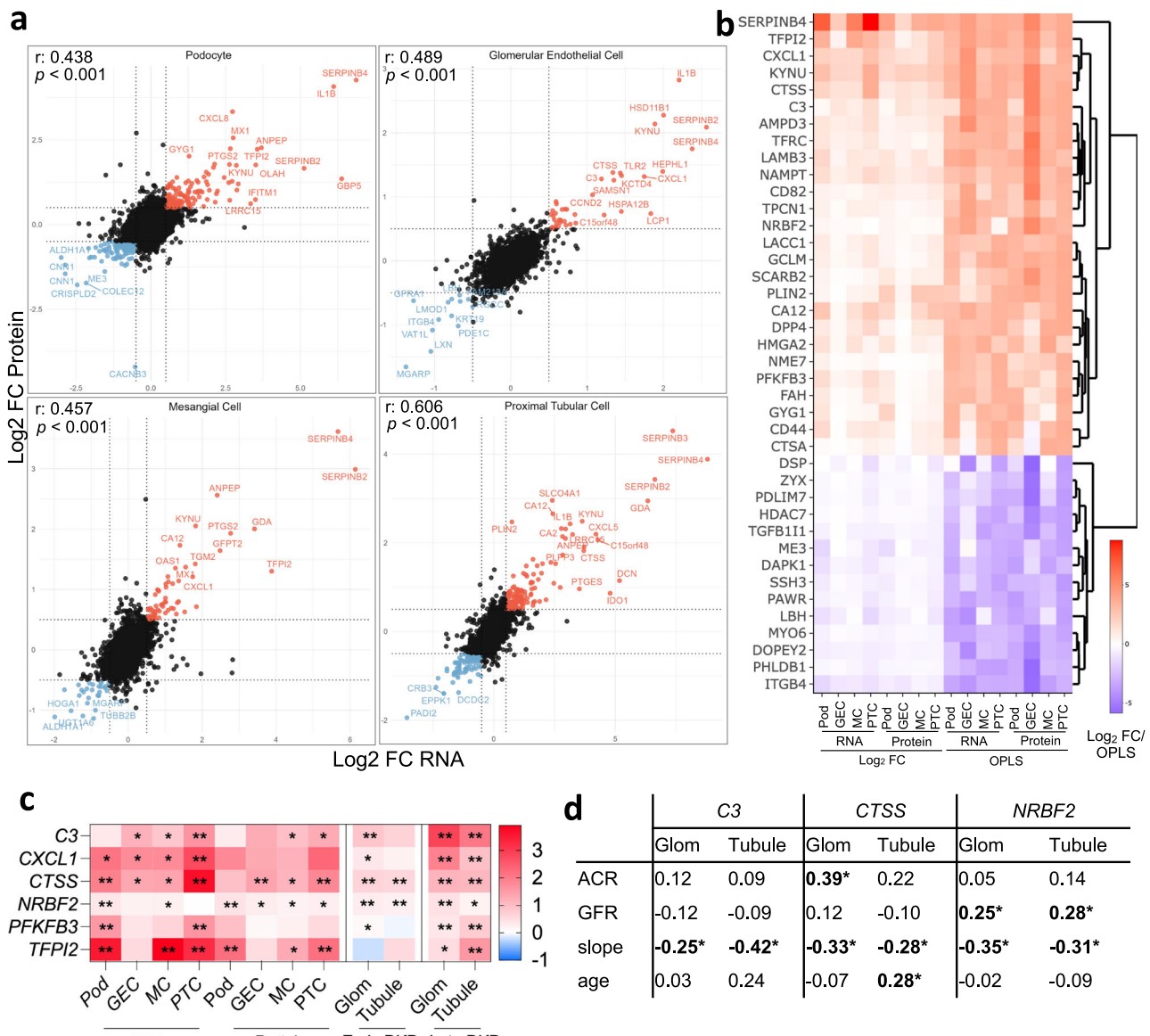

**Fig. 3 | Integrated analysis of the proteome and transcriptome highlights consistently regulated genes and proteins in insulin-resistant kidney cells and human DKD. a** Pearson correlation between transcript- and protein-level regulation in insulin-resistant vs. basal conditions, within each IR-transduced cell line ($n = 5$ biological repeats/condition). Examples of genes consistently regulated at the transcript and protein level are highlighted; differential expression analyses of transcriptomics and proteomics datasets from IR-transduced cells are available in the 'Source data' file. **b** Hierarchical clustering and heatmap of combined DE (Log₂ FC) and consensus-OPLS analysis to highlight the 'Top 40' consistently regulated proteins and transcripts across all cell lines in insulin resistant vs. basal conditions (selected if VIP > 1 and FDR < 0.1 in at least three comparisons), data are available in

'Source data' file. **c** Heat map highlighting genes with evidence of consistent regulation (Log₂ fold change) in human cell lines and either early- (American Indian type 2 diabetes cohort, glomerular 'Glom', $n = 69$ and tubular 'Tubule', $n = 47$) or late-stage DKD (ERCB cohort, 'Glom' $n = 12$ and 'Tubule' $n = 17$) vs. Living donors ($n = 18$), *FDR < 0.1, **FDR < 0.01, differential expression and significance estimated using limma, data available in 'Source data' file. **d** Correlation (Spearman R) between glomerular ('Glom', $n = 69$) and tubular ('Tubule', $n = 47$) gene expression (Log₂ mRNA intensity) and urinary albumin/creatinine ratio (ACR), glomerular filtration rate (GFR), estimated GFR decline (slope) and age in the American Indian early type-2 diabetes cohort, *$p < 0.05$.

hierarchically clustered with other similar GO terms (based on the GO structure and semantic similarity), using GOSemSim[27] (Supplementary Figs. 9–11). This highlighted clusters of consistently enriched pathways across all insulin-resistant cell lines, including the immune/inflammatory response (Fig. 5a, Supplementary Data 4), ER stress/the unfolded protein response (UPR) (Fig. 5d, Supplementary Data 5) and glycoprotein metabolism/processing (Fig. 5g, Supplementary Data 6), at both the transcript and protein level.

For each of these pathway clusters, we used the "core enrichment" transcripts/proteins from the GSEA to prioritise a subset of genes that

were driving the enrichment in all insulin-resistant kidney cell types at both the transcript and protein levels (as displayed in Fig. 5a, d and g). To evaluate whether consistent pathway enrichment was occurring in human kidneys, we calculated $Z$-scores for the expression values for these prioritised pathway genes in biopsies from early- or advanced-stage DKD and healthy living donors.

This revealed a consistent increase in pathways related to the inflammatory/immune response (Fig. 5b, c), ER stress (Fig. 5e, f), and glycoprotein processing (Fig. 5h, i), in human glomerular and/or tubular compartments of either early- or advanced-stage DKD, at the

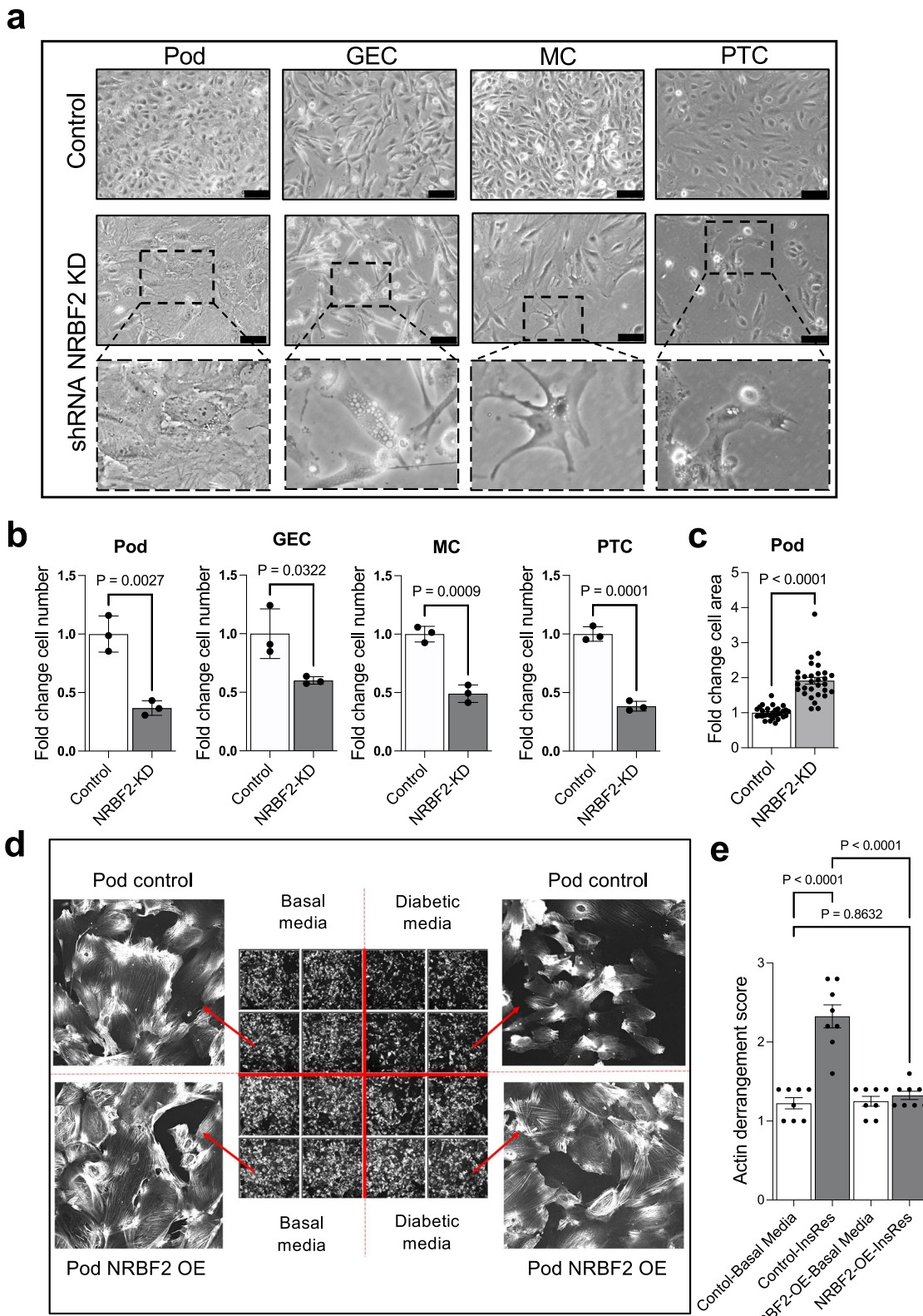

transcript level. Of note, the increases observed in the human tubular fractions were predominantly observed in individuals with advanced DKD (Fig. 5c, f, i), whilst glomerular changes were observed in both early and advanced DKD (Fig. 5b, e, h). Other examples of consistently regulated pathways identified from our insulin-resistant cell models included an increase in iron transport and a reduction in HIPPO signalling, cellular response to ammonium ion and regulation of pinocytosis (Supplementary Fig. 12).

## Insulin resistance promotes cell-type-specific changes to the proteome and the transcriptome in the kidney

To understand the heterogeneity of kidney cell responses to insulin resistance, we additionally investigated the transcripts (Supplementary Fig. 13a) and proteins (Supplementary Fig. 13b) that were regulated in a cell-type-specific manner between our cell lines.

The top cell-line-specific transcript and protein changes (selected based on both fold-change and FDR values) are presented in Fig. 6a, b

**Fig. 4 | Effect of NRBF2 knockdown and overexpression in cultured kidney cells. a** Brightfield images of podocytes (Pod), glomerular endothelial cells (GEC), mesangial cells (MC) and proximal tubular cells (PTC) showing changes in cell morphology 4 days after shRNA NRBF2 knockdown compared with scrambled shRNA controls. NRBF2 knockdown induces cell vacuolisation (enlarged images) along with podocyte hypertrophy and loss of GEC, MC and PTC. Scale bar = 100 μm. **b** Bar chart showing reduced cell number 4 days after shRNA NRBF2 knockdown compared with scrambled controls. Unpaired two-tailed *t*-test, cells were counted in three fields of view (*n* = 3 biological repeats), data are presented as mean values ± SEM. **c** Bar chart showing increased cell area in shRNA NRBF2 knockdown podocytes compared with controls. Area was measured in 10 cells in each of 3 fields of view. Unpaired two-tailed *t*-test, (*n* = 3 biological repeats), data are presented as mean values ± SEM. **d** Images of phalloidin-stained podocytes overexpressing *NRBF2* (pod NRBF2 OE) and wild-type controls cultured for 10 days in basal or diabetic media ('DM'). Diabetic media-induced changes in cell morphology and *F*-actin distribution (top right) that are attenuated by NRBF2 overexpression (bottom right). **e** Quantification of *F*-actin stress fibres, indicating significant *F*-actin rearrangement in wild-type podocytes exposed to Diabetic media 'DM' and no difference in *F*-actin distribution in NRBF2-overexpressing podocytes, one-way ANOVA with Tukey's multiple comparisons test (*n* = 8 technical repeats), data are presented as mean values ± SEM.

and include several long-non-coding RNAs. Notable KEGG pathways that were enriched in a cell-type-specific manner in response to insulin resistance included RIG-I-like-receptor signalling in Pods; glycosaminoglycan degradation in GECs; ECM-receptor interaction, DNA replication and the cell cycle in MCs; and Complement/coagulation cascades and a reduction of tight junctions in PTCs (Supplementary Fig. 13c). Although limited to genes where we had information on both transcript and protein abundance, we also identified several protein-coding genes with evidence of cell-type-specific regulation at both the transcript and protein level in response to insulin resistance (Fig. 6c).

**Evaluation of prioritised genes in human DKD using single-cell sequencing data.** Cell-line-specific responses to insulin resistance were further evaluated in human kidney biopsy single-cell sequencing data. We first mapped the single-cell profiles of our cell-line-specific insulin-resistance genes, which indicated that a subset displayed cell-type-specific expression patterns between Pod, ECs, MCs and PTCs (i.e., our cell types of interest) in vivo, in healthy human kidney (Supplementary Fig. 14a, b). We further prioritised the genes that were displaying evidence of consistent cell-type-specific regulation in human DKD, aligning our in vitro cell type data with single-cell sequencing data from an American Indian type 2 diabetes cohort[28] (early DKD) and the kidney precision medicine project (KPMP)[29] (advanced DKD) (Supplementary Figs. 15 and 16).

This highlighted genes that were also consistently, specifically, regulated in corresponding single-cell types from kidney biopsies as well as our in vitro studies. These included *TCF21*, encoding 'Transcription factor 21' and *RASL11B* encoding 'Ras-like protein family member 11' in Pods (Fig. 6d, h); MGP, encoding 'Matrix gla Protein' in GECs (Fig. 6e, i); *TWF2*, encoding 'Twinfilin-2' in MCs (Fig. 6f, j); and *BDH2*, encoding '3-hydroxybutarate dehydrogenase 2', *S100A1*, encoding 'S100 calcium-binding protein A1' and *PIGR*, encoding 'Polymeric immunoglobulin receptor' in PTCs (Fig. 6g, k), in either dataset. This provides further evidence that cell-specific responses to insulin resistance occur in human DKD and highlights additional genes for mechanistic follow-up studies.

**Insulin-resistant kidney cells have differential, protein-level regulation of mitochondrial dynamics**

Further investigation of the molecular pathways that were differentially regulated between our 4 insulin-resistant kidney cell types indicated a dramatic cell-type-specific disruption to pathways governing mitochondrial dynamics, which was uniquely observed at the protein level (Fig. 7a); highlighting an important cell-type-specific response to insulin resistance, not captured by transcriptomics data alone.

We observed a significant reduction in mitochondrial bioenergetic processes ('oxidative phosphorylation', OXPHOS, and the 'electron transport chain', ETC), specifically at the protein level in insulin-resistant Pods, MCs and PTCs, alongside a reduction in pathways regulating mitochondrial gene expression (Fig. 7a). In insulin resistant GECs, however, there was a significant positive enrichment of proteins involved in mitochondrial transcription and translation.

The core proteins driving these enrichment results included proteins involved in mitochondrial gene expression, mitochondrial protein assembly, the TCA cycle, as well as several subunits of complex I of the respiratory chain (Supplementary Data 7). Overall, 651 mitochondrial proteins were detected across all cell types (annotated using GOCC), of which 98 were downregulated in insulin-resistant Pods (FDR < 0.1) (Fig. 7b). Many of these proteins were also downregulated in MCs (18 proteins, FDR < 0.1) and PTCs (31 proteins, FDR < 0.1).

Interestingly, of the 27 complex I subunits that we detected at both the transcript and protein level, 18 (66.7%) were significantly (FDR < 0.1) reduced in Pods and 3 in PTCs. Furthermore, 5 members of complex IV proteins were significantly reduced in podocytes (Fig. 7c, FDR < 0.1). Several were also significantly regulated in PTCs at the nominal *p* value threshold (0.05) but did not pass the multiple correction threshold (Supplementary Fig. 17a). Protein subunits of ETC complex IV were also highly downregulated in insulin-resistant Pods (Fig. 7c, Supplementary Fig. 17a). In contrast, we found no evidence for the regulation of any respiratory complexes in GECs (Fig. 7c, Supplementary Fig. 17a). Validation experiments using qPCR and western blotting demonstrated that, in all the cell types, there was no significant difference in the mRNA levels of any of the OXPHOS subunits studied in insulin resistant vs. insulin sensitive conditions (Fig. 7g, i, Supplementary Fig. 17b). In contrast, we found a reduction in protein expression of Complex IV subunits (CoxII) in pods, MC and PTC (Fig. 7h, Supplementary Fig. 17c), as well as Complex I (NDUFB8) in MCs (Fig. 7j, Supplementary Fig. 17c). In GECs, we found no evidence of regulation of any of the protein complex subunits studied (Fig. 7h, j, Supplementary Fig. 17c).

The discordance between transcript and protein regulation was also evident for many of the enzymes involved in the TCA cycle, which excluding Aconitase 1 (ACO1), were again largely down-regulated at the protein level in Pods, MCs and PTCs (Fig. 7d, Supplementary Fig. 17d). Interestingly, a post-transcriptional down-regulation of proteins involved in glycolysis was not observed in any of the insulin resistant cell types (Fig. 7e, Supplementary Fig. 17e), potentially indicating that insulin-resistance could promote a "glycolytic switch" in Pods, MCs and PTCs.

**Mitochondrial bioenergetics are selectively impaired in insulin-resistant kidney cells**

To further assess whether insulin resistance resulted in functional cell-line-specific disruption of mitochondria (particularly mitochondrial bioenergetics), we performed live cell analysis using a 'Seahorse XF Analyzer'. Oxygen consumption rate (OCR) and Extracellular acidification rate (ECAR) were continuously monitored in basal and insulin-resistant cells, sequentially incubated with oligomycin, FCCP and antimycin A plus rotenone (Fig. 8a). Results of these experiments demonstrated that insulin resistant Pods, MCs and PTCs had reduced mitochondrial ATP production (Fig. 8b), whereas ATP production via glycolysis was largely maintained or, in the case of Pods and PTCs, enhanced (Fig. 8c). However, glycolysis was not able to fully

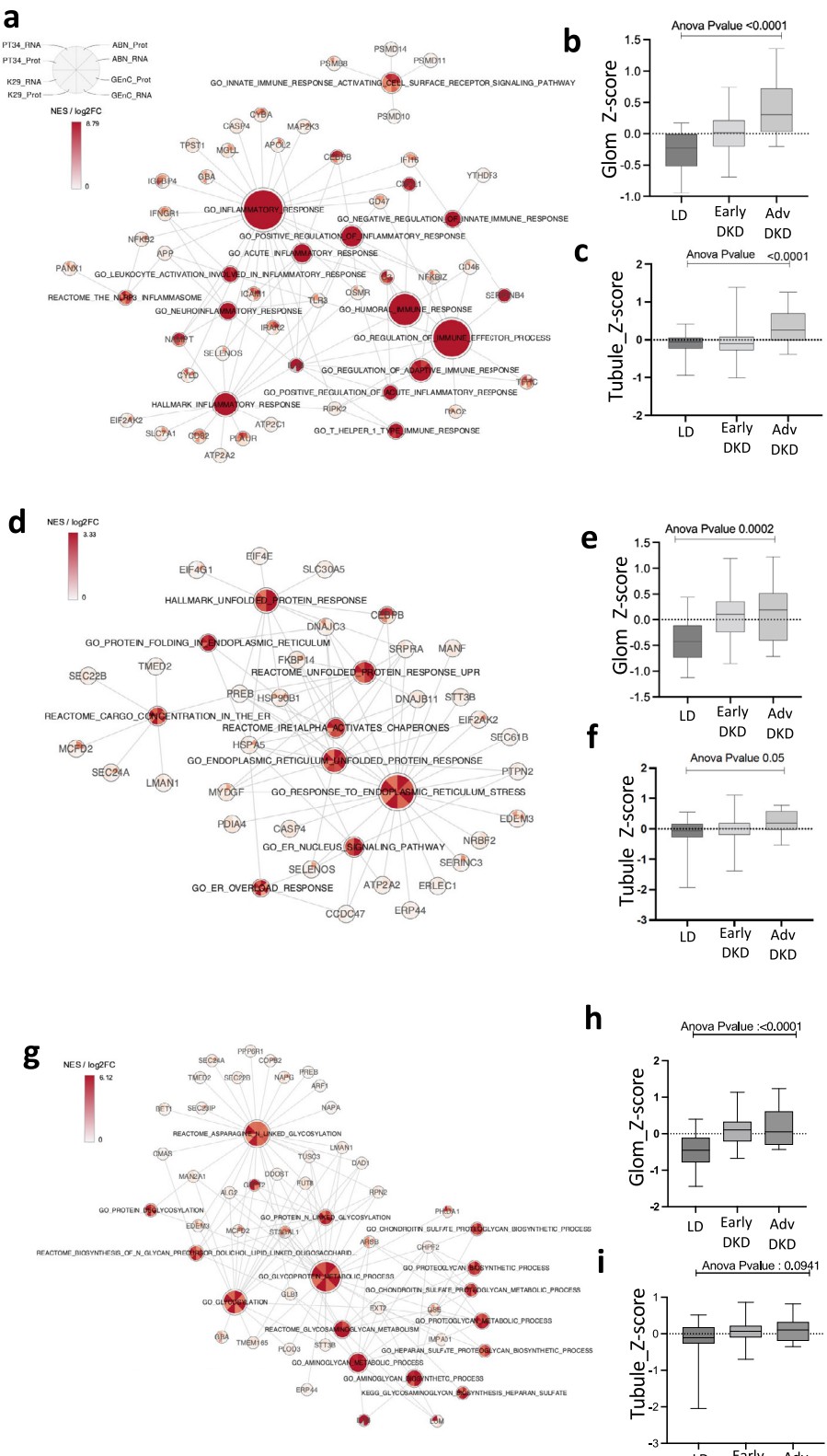

compensate for the loss of mitochondrial respiration, as we observed an overall reduction in ATP production rate in insulin resistant Pods, MCs and PTCs (Supplementary Fig. 18a).

In insulin-resistant Pods, MCs and PTCs, maximal respiration and mitochondrial spare capacity (or 'mitochondrial reserve', reflecting healthy mitochondria[30]) were also reduced (Fig. 8d, e, Supplementary Fig. 18b), aligning with the reduction in mitochondrial proteins detected in these cells. No significant changes to any of the cellular

bioenergetic parameters studied were apparent in insulin-resistant GECs (Fig. 8).

Collectively, these results demonstrate that insulin resistance is associated with a cell-line-specific regulation of mitochondrial pathways. Given the wide-ranging effects of mitochondria on cell function and cell signalling (beyond bioenergetics), further comprehensive studies of the effects of insulin resistance on mitochondrial functional parameters are warranted.

**Fig. 5 | Insulin-resistant kidney cells are characterised by an increased inflammatory response, ER stress and glycoprotein metabolism pathways. a** 'Gene-concept network' displaying normalised enrichment scores (NES) for immune/inflammatory response pathways enriched in at least one cell type at RNA and protein level ($p$-value < 0.05, $q$-value < 0.1 either from DE or Consensus OPLS analysis) and Log$_2$ Fold Change values of core enrichment inflammatory genes/proteins, consistently regulated in each insulin resistant cell line. **b, c** Box plot displaying average $Z$-scores of expression for core inflammatory/immune genes in **b** human glomerular and **c** tubular bulk transcriptomics data from both early (American Indian type-2 diabetes cohort, glomerular ('Glom'), $n = 69$ and tubular ('Tubule'), $n = 47$) and advanced-stage DKD (ERCB cohort, 'Glom', $n = 12$ and 'Tubule' $n = 17$) vs Living donors ($n = 18$). **d** 'Gene-concept network' displaying NES for ER stress pathways enriched in at least one cell type at RNA and protein level ($p$-value < 0.05, $q$-value < 0.1 from DE or Consensus OPLS analysis) and Log$_2$ Fold Change values of core enrichment ER stress genes/proteins, consistently regulated in each insulin resistant cell line. **e, f** Box plots displaying average $Z$-scores of expression for core ER stress genes in **e** human glomerular and **f** tubular bulk transcriptomics data from both early ('Glom', $n = 69$ and 'Tubule', $n = 47$) and advanced-stage DKD ('Glom', $n = 12$ and 'Tubule' $n = 17$) vs. Living donors ($n = 18$). **g** 'Gene-concept network' displaying NES for glycoprotein biosynthesis/metabolism pathways enriched in at least one cell type at RNA and protein level ($p$-value < 0.05, $q$-value < 0.1 from DE or consensus OPLS analysis) and Log$_2$ Fold Change values of core enrichment glycoprotein biosynthesis/metabolism genes/proteins, consistently regulated in each insulin resistant cell line. **h, i** Box plots displaying average $Z$-scores of expression for core glycoprotein-related genes in **h** human glomerular and **i** tubular bulk transcriptomics data from both early ('Glom', $n = 69$ and 'Tubule', $n = 47$) and advanced-stage DKD ('Glom', $n = 12$ and 'Tubule' $n = 17$) vs. Living donors ($n = 18$). For **b, c, e, f, h** and **i** one-way ANOVA shown.

## Discussion

Insulin resistance has been highlighted as a key metabolic determinant of DKD, in both type 1[7,31] and type 2[8] diabetes. Furthermore, it is now clear that insulin resistance drives kidney disease before, during, and after diabetes onset, across ancestries[5,9,32,33]. In this study, we modelled insulin sensitivity and insulin resistance in human kidney cells known to be affected early in DKD progression, compared with human early- and advanced-stage DKD biopsy data, and provided an overview of the consistent and cell-type-specific molecular changes associated with insulin resistance. Although the use of in vitro models cannot fully represent the complex changes observed in vivo (and, indeed, we focus on insulin resistance as an important molecular driver of DKD), we demonstrate the value of these models for the manipulation of target genes and pathways of interest, to provide further mechanistic insights.

Our integrated analysis of the proteome and the transcriptome identified a set of 40 transcripts and proteins that were commonly regulated in all insulin-resistant kidney cells studied. Targeted analyses of human transcriptome data from micro-dissected glomeruli and tubules demonstrated a consistent (up)regulation in individuals with DKD, of genes with both known roles in the kidney (*CTSS*[34], *C3*[35–37], *CXCL1*[38], *PFKFB3*[39,40], *TFPI2*[41]) and with no prior direct links to kidney function (*NRBF2*); prioritising important molecular targets for further detailed studies of DKD. As a 'proof-of-concept' to show the value of cell models in both informing targeted analysis of human population data and for follow-up studies, we subsequently investigated the mechanistic role of *NRBF2* in our models. Our in vitro studies demonstrated that *NRBF2* knock-down had detrimental effects in Pods, GECs, MCs and PTCs and, in contrast, indicated that increased *NRBF2* expression in podocytes protected against actin remodelling induced by insulin resistance. Collectively, these results point towards an important functional role of *NRBF2* in the kidney and indicate the importance of controlled *NRBF2* expression levels in kidney cells. These results also suggest the functional consequences of aberrant *NRBF2* expression likely differ between cell types. Although NRBF2 has not previously been studied in the context of DKD, this protein is emerging as an important regulator of mTOR-mediated autophagy[42,43]. This role in regulating autophagy is plausibly the mechanism by which NRBF2 exerts its effects on kidney cells and is an important area of future investigation.

Regarding genes with prior evidence for a role in kidney function, we found that increased kidney *C3* and *CTSS* expression was associated with eGFR decline and/or albuminuria. Of note, systemic inhibition of Cathepsin S protein (CTSS) has previously been shown to protect against albuminuria and glomerulosclerosis in diabetes[34]. Taken with our results, this suggests a local upregulation of *CTSS* expression (which may be driven by insulin resistance) occurs in the glomeruli and tubules in DKD which may contribute towards kidney damage. Likewise, blockade of the receptor for C3a (a cleavage product of C3 in the

activation of both the classical and alternative complement pathways), has been found to have protective effects in animal models of DKD[36,37]. Our results provide further support for the involvement of C3 (and, therefore, the complement system) in DKD and evidence for local production and regulation of *C3* by resident kidney cells.

Our comprehensive gene set enrichment analysis, integrating protein and transcript level data, also highlighted several common biological pathways that were dysregulated across all insulin-resistant kidney cell types studied and were similarly regulated in kidney biopsies from individuals with DKD. These included ER stress and inflammatory/immune pathways, in addition to glycoprotein processing[44]. Although it is unsurprising that glycoprotein synthesis is dysregulated in insulin-resistant conditions, given that metabolic disturbances can increase the complexity of glycan branching and altered plasma N-glycosylation patterns are associated with DKD progression[45], the specific glycoprotein changes in the kidney in DKD, and their functional consequences, are not fully characterised. Given that the major roles for glycoproteins in the kidney include the formation of the glomerular basement membrane (GBM)[46], the glomerular endothelial glycocalyx[47] and the podocyte slit diaphragm[48], comprehensive profiling of glycoprotein composition in kidney cells in diabetes is clearly warranted. Our results here provide examples of core glycoprotein-modifying proteins (including several enzymes) that are upregulated in insulin-resistant kidney cells and in human DKD; which may be targets to restore kidney glycoproteins in DKD. Similarly, although both ER stress and inflammatory responses are commonly linked to DKD pathogenesis[49,50], our results provide examples of genes and proteins that may be key in driving ER stress and immune/inflammatory responses in insulin-resistant Pods, GEC, MCs and PTCs, and in human DKD in both glomeruli and tubules. The identification of cellular pathways that are commonly dysregulated in multiple cell types involved in DKD pathogenesis also offers the opportunity to identify processes that can be targeted by broad-acting pharmaceuticals.

In addition to the common genes and pathways dysregulated across all insulin-resistant cell types, we also explored any cell-type specific changes occurring in our in vitro models, which we further evaluated using single-cell sequencing data from early- and advanced-stage human DKD; thereby identifying important molecular changes which may be targets for cell-type-specific therapeutic strategies[51] and the focus of future mechanistic work. This identified several cell-type-specific changes in protein-coding genes, with potential mechanistic roles in DKD (although, notably, the expression of these genes was not necessarily found to be cell-type-specific). For example, Podocyte-specific changes included a reduction in *TCF21*, which encodes a transcription factor previously identified as crucial for podocyte development and maintenance. Studies using animal models have revealed that a podocyte-specific reduction in *Tcf21* promotes both Focal Segmental glomerulosclerosis (FSGS)-like disease and an

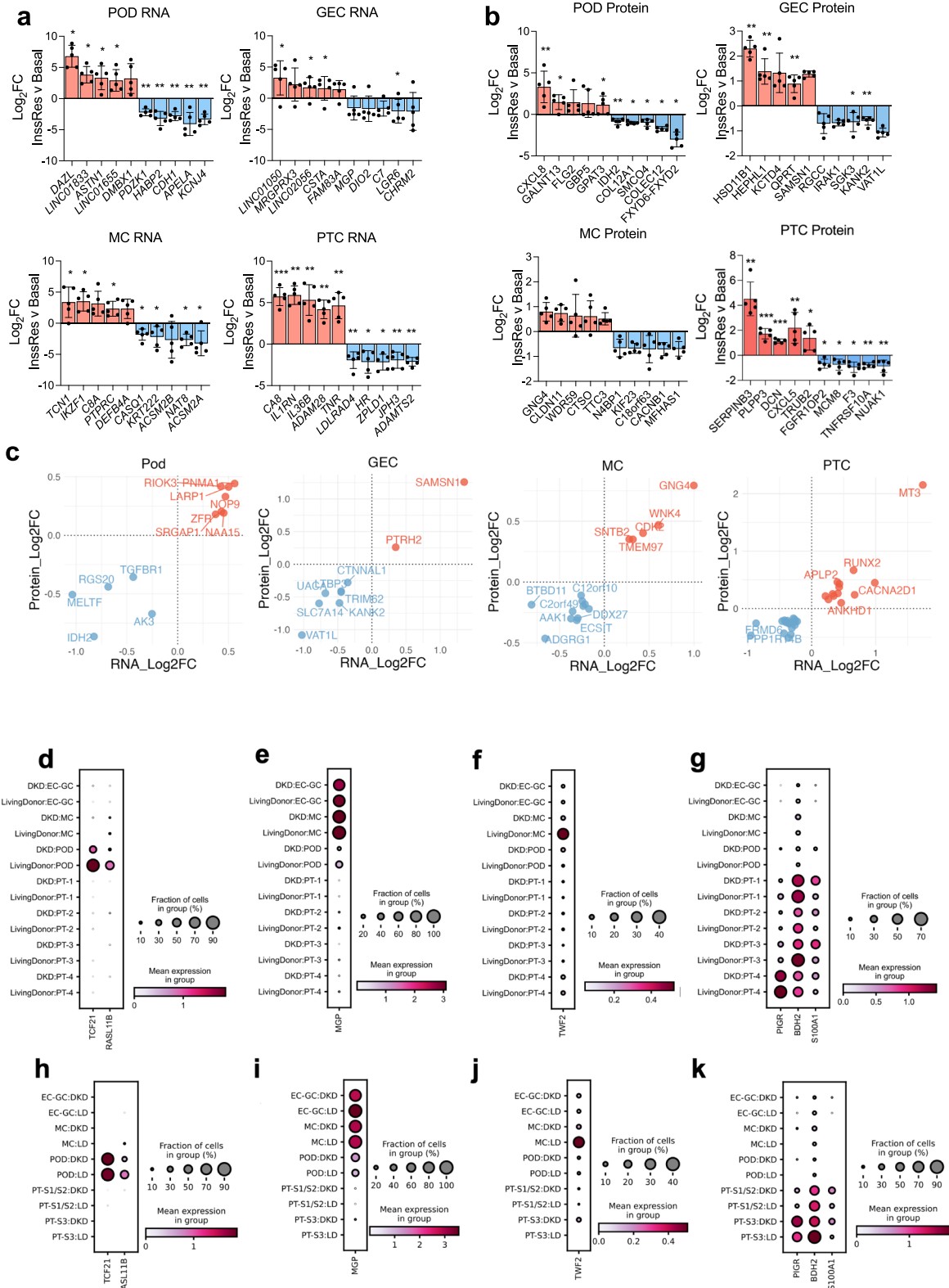

**Fig. 6 | Cell-type-specific responses to insulin resistance and targeted analysis of human kidney single cell sequencing data to identify replicated changes in human DKD. a** and **b** Bar charts demonstrating the log₂ fold-change values for the top 10 selectively regulated **a** transcripts and **b** proteins in response to insulin resistance, calculated from transcriptome and proteome data normalised and analysed for each individual cell line separately *FDR < 0.1, **FDR < 0.05, differential expression and significance estimated using limma, with a global benjamini-hochberg correction, *n* = 5 biological replicates, per

cell type, data are presented as mean values ± SEM. **c** scatter plots demonstrating the genes regulated in response to insulin resistance in a cell-type-specific manner, consistently at the transcript and protein level. **d**–**k** single-cell sequencing analysis of target genes in each cell-type cluster (dot plots displaying the percentage of expressing cells and mean expression values) in **d**–**g** an American Indian type-2 diabetes cohort with early-DKD (*n* = 44 early-DKD vs. *n* = 18 LD) and **h**–**k** advanced DKD from KPMP (*n* = 10 advanced-DKD vs. *n* = 18 living donor).

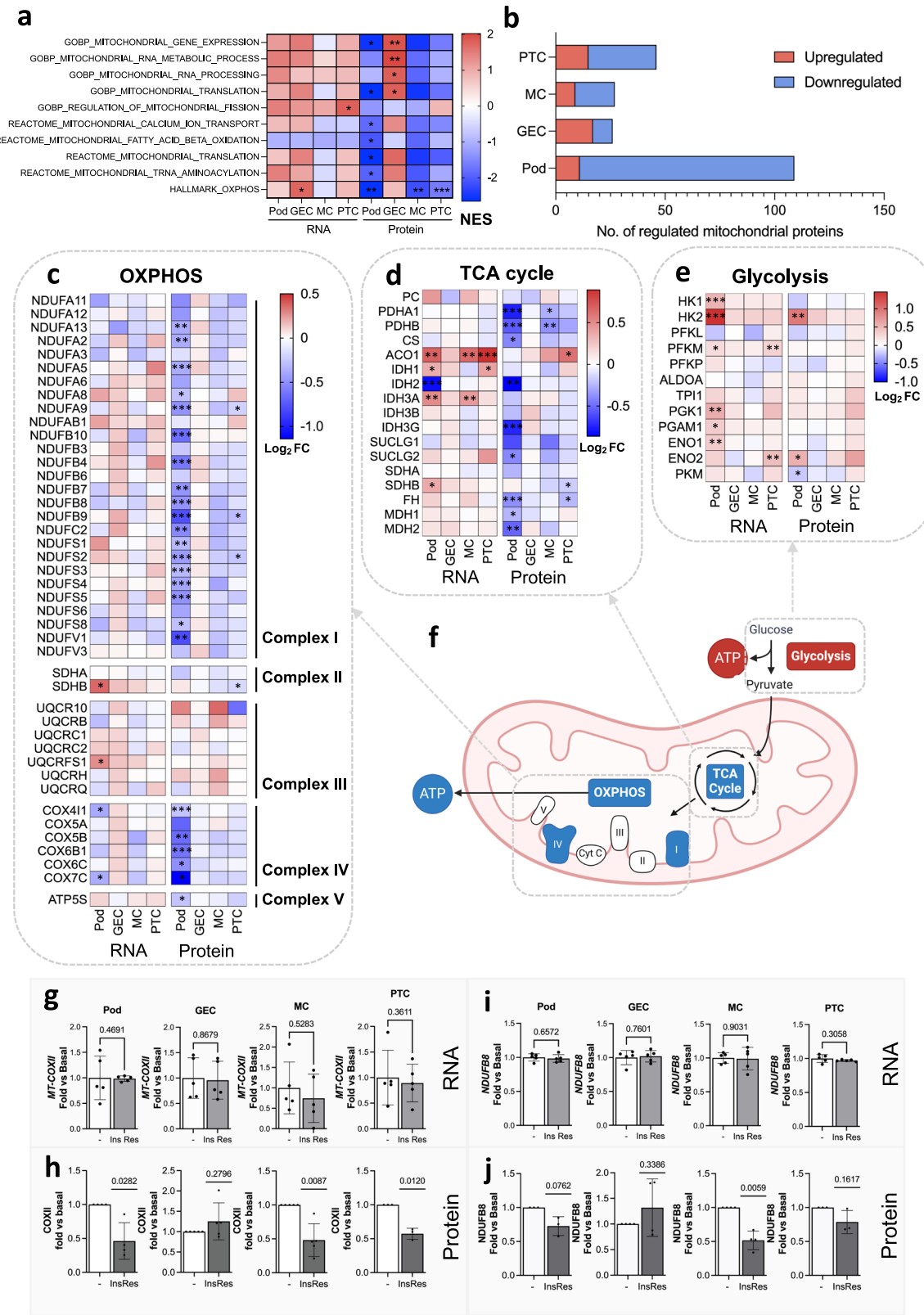

exaggerated DKD-like phenotype[52]. Our results provide further evidence that *TCF21* regulation in podocytes may play an important role in DKD pathogenesis. We also found a reduction of *RASL11B* in insulin-resistant podocytes that was present in both early- and advanced-stage DKD kidney biopsies. This small GTPase has previously been shown to be important in the accumulation of matrix in cartlidge[53], but has not been explored in the kidney. Its reduction in insulin resistance could

conceivably be a feedback response to limit matrix accumulation in this setting. PTC-specific changes were also observed for genes with prior links to PTC damage; for example, *PIGR*, encoding the polymeric immunoglobulin receptor, which has been shown to increase in PTCs in a variety of kidney diseases, including DKD[54]; and those with no known associations to PTC function, for example, *BDH2*, encoding the enzyme 3-hydroxybutarate dehydrogenase 2, involved in fatty acid

**Fig. 7 | Insulin-resistant kidney cells have differential, protein-level regulation of mitochondrial dynamics. a** Normalised enrichment scores (NES) for enriched mitochondrial gene signatures in insulin-resistant cell lines, highlighting predominant regulation at the protein-level (*$q < 0.1$, **$q < 0.05$, ***$q < 0.01$). **b** Number of significantly up- or down-regulated (FDR < 0.1) mitochondrial proteins (based on having mitochondrial GOCC annotation) detected in our proteomics datasets. **c** Heatmap of $\log_2$ Fold Change (insulin resistant vs. basal) for the respiratory chain complex transcripts and proteins detected in all four cell types *FDR < 0.1, **FDR < 0.05, ***FDR < 0.01. **d** Heatmap of $\log_2$ Fold Change (insulin resistant vs basal) for TCA cycle transcripts and proteins detected in all four cell types *FDR < 0.1, **FDR < 0.05, ***FDR < 0.01. **e** Heatmap of $\log_2$ Fold Change (insulin resistant vs. basal) for glycolysis transcripts and proteins detected in all 4 cell types *FDR < 0.1,

**FDR < 0.05, ***FDR < 0.01. **f** Schematic diagram of mitochondrial bioenergetic processes likely dysregulated in Pods, MCs and PTCs based on proteomics data (upregulated = red, downregulated = blue), created in BioRender. Sinton, M. (2023) BioRender.com/s01v107. **g** qPCR results of *MT-COXII* mRNA ($n = 5$, each cell type and condition, two-tailed $t$-test, data are presented as mean values ± SEM) and **h** densitometry values for COX-II protein expression ($n = 4$ podocyte, $n = 5$ glomerular endothelial cells, $n = 5$ mesangial cells, $n = 3$ proximal tubular cells, two-tailed $t$-test, data are presented as mean values ± SEM), **i** qPCR results of *NDUFB8* mRNA ($n = 5$, each cell type and condition, two-tailed $t$-test) and **j** densitometry values for NDUFB8 protein expression ($n = 3$ podocyte, $n = 4$ glomerular endothelial cells, $n = 4$ mesangial cells, $n = 3$ proximal tubular cells, two-tailed $t$-test).

beta oxidation and epithelial cell differentiation; and *S100A1*, a member of the S100 family of calcium binding proteins with an array of downstream molecular functions. Of note, S100A1 has attracted particular interest as a therapeutic target for cardiovascular diseases including heart failure[55,56], as such, further studies investigating the effects of this gene in PTCs are clearly justified. In MCs, we found a significant reduction in *TWF2*, encoding the actin-regulating protein Twinfilin-2 in both early- and advanced-stage DKD, the role and implications of which remain to be explored. GECs displayed high basal expression of *MGP*, a vitamin K-dependent protein that is a potent inhibitor of vascular calcification[57]. Although *MGP* was also detected in Pods and MCs in our human biopsy data, the reduction in *MGP* expression in DKD was uniquely observed in GECs in both early and, more profoundly, advanced-stage DKD biopsy samples. This loss of *MGP* in GECs in DKD could contribute towards calcification. Further examples of cell-type-specific responses to insulin resistance included the differential expression of multiple long-non-coding RNAs (lncRNAs), which were among the top cell-line-specific transcriptional changes.

Our pathway analysis demonstrated that mitochondrial dynamics and bioenergetics were selectively impaired between different insulin-resistant kidney cell types and that this was uniquely observed at the protein level. These results highlight the importance of post-transcriptional regulation of mitochondrial metabolic proteins (for example, impaired translation and/or increased protein breakdown, including mitophagy), indicating an important mechanism of bioenergetic regulation that may not be accurately captured in (or interpreted from) transcriptomics data alone. Indeed, post-transcriptional regulation of mitochondrial bioenergetic function has been recently described in type-2 diabetic islets[58]. Although there is a well-acknowledged association between insulin resistance and mitochondrial dysfunction[59] and the importance of disturbances to mitochondrial function in the pathogenesis of kidney disease (including mitochondrial degradation in glomeruli and tubules in CKD[60]) is increasingly appreciated[61–63], the role of insulin resistance in promoting mitochondrial dysfunction in the kidney and the comparison of mitochondrial adaptations in insulin resistance between kidney cell types is less well-defined. In other tissues, cellular insulin signalling has been shown to directly regulate mitochondrial metabolism[64] and a loss of insulin signalling (via IR or combined IR/IGF-IR) in muscle can reduce mitochondrial respiration via complex I[65], although this is likely a cell-type- and context-specific effect. Previous studies inhibiting IR/IGF-IR signalling in podocytes have, in fact, shown protective effects against severe mitochondrial dysfunction caused by the deletion of *Phb2*[64].

Our results indicate that insulin resistance in Pods, MCs and PTCs is associated with a reduction of several mitochondrial proteins (including multiple subunits of the ETC protein complexes) and a subsequent shift to glycolysis being the predominant pathway by which cellular ATP is produced. However, in each case, glycolysis was not sufficient to compensate for reduced mitochondrial ATP

production and an overall suppression of ATP production was observed in insulin-resistant Pods, MCs and PTCs. A generalised mitochondrial dysfunction was also clearly observed in insulin-resistant Pods, MCs, and PTCs indicated by reduced spare respiratory capacity and maximal respiration. This likely reflects the substantial reduction in ETC protein subunits (particularly in podocytes) in combination with a reduction of TCA proteins, including the pyruvate dehydrogenase subunits PDHA1 and PHDB (thereby limiting pyruvate metabolism). Indeed, loss of pyruvate dehydrogenase activity is sufficient to reduce spare capacity[30,66]. Since ETC and TCA proteins reside in the mitochondria, this result may also reflect generalised reduction in functional mitochondria in insulin-resistant Pods, MCs and PTCs. In contrast, insulin-resistant GECs displayed an enrichment for proteins involved in mitochondrial biogenesis and no evidence of reduced mitochondrial bioenergetic function. This could imply an effective removal (e.g., via mitophagy) and replacement (e.g., via increased mitochondrial biogenesis) of defective mitochondria in these cells, which requires further investigation. Although, in general, endothelial cells are regarded as having a comparatively low energy demand[67]. Future in-depth, cell-type-specific pathway enrichment and thorough integration with single-cell multi-omics data from human cohorts (as well as comparison with results from other experimental approaches) will no doubt identify additional cell-type-specific responses to insulin resistance in DKD; thereby highlighting future opportunities for cell-type-specific therapeutic targeting (e.g., for example, with AAV-delivered gene therapy[51]).

We recognise there are constraints to this study. This work aimed to explore the molecular changes related to insulin resistance, in kidney cell types that are known to be important in the early stages of DKD pathogenesis. We used established in vitro cell lines to model insulin sensitivity and insulin resistance and provide a comprehensive overview of the molecular changes. Although there are well-recognised limitations in the use of cell lines as model systems, such as losing cellular markers over time and, of course, in vitro models cannot fully capture the complex in vivo scenario, our dataset provides a proof-of-principle for the use of these models to identify targets for focused validation analysis in human population data and as a platform for follow-up studies, to provide further mechanistic insight. Of note, we also focus our model on insulin resistance as an important molecular mediator of DKD and therefore, may not capture or represent expression changes that are caused by other drivers of this complex disease. Our validation in human DKD cohorts uses kidney single-cell sequencing and bulk glomerular and tubular transcriptomics data from early-stage and late-stage DKD. We were unable to confirm protein changes in our biopsy specimens. In our analysis of single-cell sequencing data, we highlighted examples of genes that are regulated in a cell-type-specific manner in human DKD, consistent with cell-line-specific regulation in response to insulin resistance in vitro. However, these example genes are not necessarily expressed in a cell-type-specific manner, which will need to be considered in the design of any future follow-up studies. While some of our analyses in human cohorts

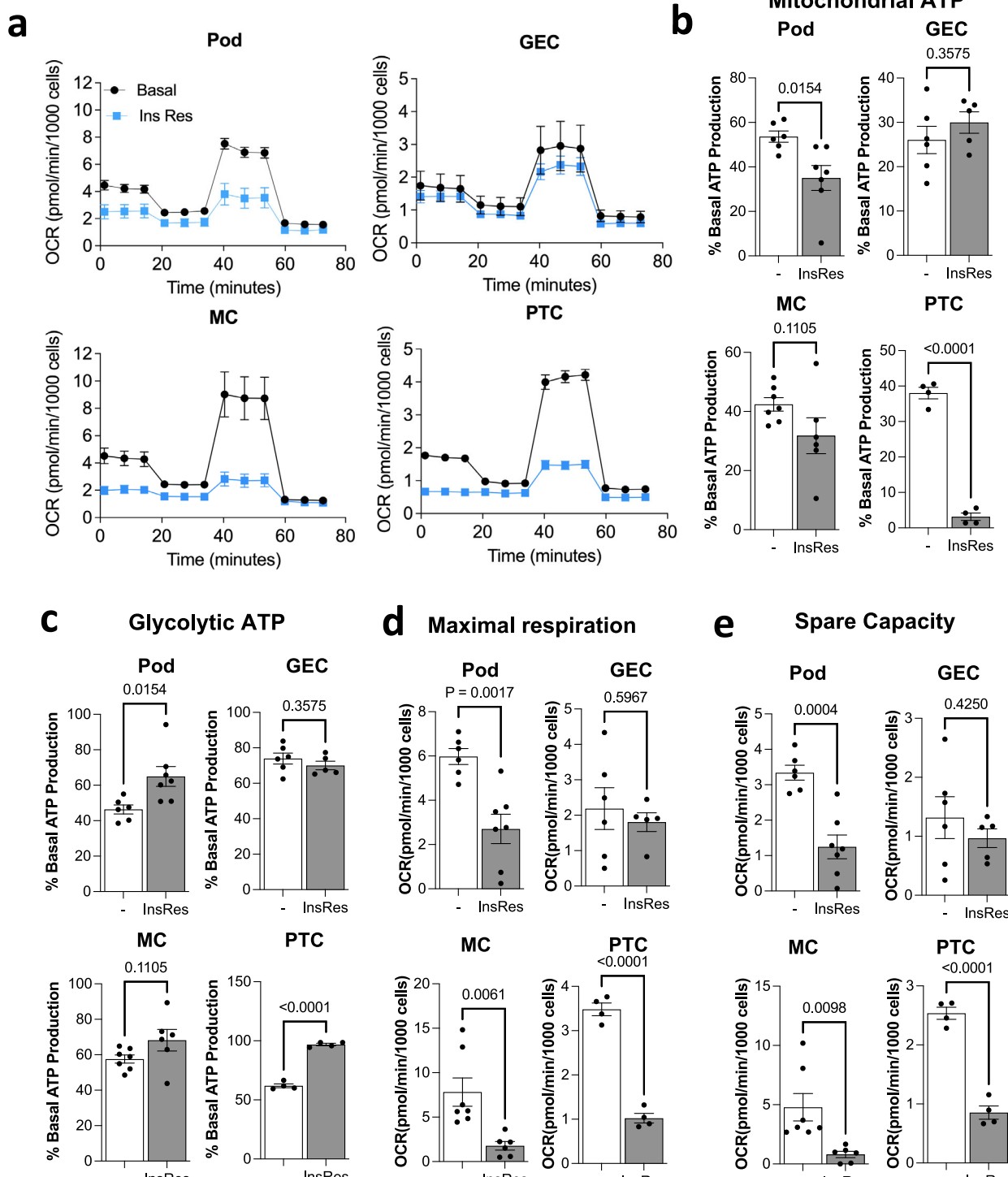

**Fig. 8 | Mitochondrial metabolism is differentially impaired in insulin-resistant kidney cells. a** Seahorse extracellular flux analysis of oxygen consumption rate (OCR) in 'Basal' or 'Insulin resistant' podocytes, glomerular endothelial cells (GEC), mesangial cells (MCs) and proximal tubular cells (PTCs) following injections of oligomycin (1.5 μM), FCCP (1 μM) and antimycin A (0.5 μM) plus rotenone (0.5 μM) at the indicated time points. **b**, **c** Percentage of ATP production attributed to **b** mitochondrial respiration or **c** glycolysis in each cell line under 'Basal' or 'Insulin Resistant' conditions in podocytes ('Pod', n = 6 'Basal' vs. n = 7 'Insulin Resistant'), glomerular endothelial cells ('GEC', n = 6 'Basal' vs. n = 5 'Insulin Resistant'), mesangial cells ('MC', n = 7 'Basal' vs. n = 6 'Insulin Resistant') and proximal tubular cells (PTC, n = 4 'Basal' vs. n = 4 'Insulin Resistant') two-tailed t-test, data are presented as mean values ± SEM; **d** OCR values per minute, per 1000 cells representing maximal respiratory capacity (in 'Insulin Resistant' vs. 'Basal' conditions in Podocytes ('Pod', n = 6 'Basal' vs. n = 7 'Insulin Resistant'), glomerular endothelial cells ('GEC', n = 6 'Basal' vs. n = 5 'Insulin Resistant'), mesangial cells ('MC', n = 7 'Basal' vs. n = 6 'Insulin Resistant') and proximal tubular cells (PTC, n = 4 'Basal' vs. n = 4 'Insulin Resistant') two-tailed t-test, data are presented as mean values ± SEM; **e** OCR values per minute, per 1000 cells representing mitochondrial spare respiratory capacity (in 'Insulin Resistant' vs. 'Basal' conditions in podocytes ('Pod', n = 6 'Basal' vs. n = 7 'Insulin Resistant'), glomerular endothelial cells ('GEC', n = 6 'Basal' vs. n = 5 'Insulin Resistant'), mesangial cells ('MC', n = 7 'Basal' vs. n = 6 'Insulin Resistant') and proximal tubular cells (PTC, n = 4 'Basal' vs. n = 4 'Insulin Resistant') two-tailed t-test, data are presented as mean values ± SEM.

used information from different datasets (which may introduce variation due to technical differences), batch-to-batch variation was minimised through harmonisation of study protocols, including for sample collection. Appropriate batch correction methods were also applied[68].

In summary, by providing a comprehensive overview of kidney cell responses to insulin resistance, our results provide further insights into the molecular changes underlying insulin-resistance-driven DKD and highlight potential therapeutic targets for future study. Our focused investigation of early and late-stage human biopsy data confirmed consistent regulation of multiple "insulin-resistance-associated genes" in human DKD, including *C3, CXCL1, CTSS, NRBF2, PFKFB3* and *TFPI2*. We provide further evidence that kidney inflammation, ER stress and glycoprotein metabolism are enhanced in DKD, which may be driven by cellular insulin resistance and highlight a previously under-appreciated discordance in the regulation of mitochondrial proteins vs. transcripts in the kidney. Furthermore, our data provide an important resource for researchers to investigate their molecules of interest and to assess the utility of these cell models as molecular tools when designing future mechanistic studies.

## Methods

### Ethics
For the samples analysed from the American Indian type-2 diabetes cohort, each participant signed an informed consent document, and the study was approved by the Institutional Review Board of the National Institute of Diabetes and Digestive and Kidney Diseases (NIDDK). Biopsy samples in ERCB were obtained from patients after informed consent and with the approval of the local ethics committees.

### Cell culture and conditions
Conditionally immortalised human podocytes[20] and mesangial cells[22] were maintained in RPMI-1640 containing L-glutamine and NaHCO$_3$, supplemented with 10% or 20% FBS, respectively (Gibco). Conditionally immortalised human glomerular endothelial cells[21] were maintained in endothelial cell growth medium-2, containing microvascular SingleQuots Supplement Pack in 5% FBS (Lonza). Conditionally immortalised human proximal tubular cells[23] were maintained in DMEM-HAM F-12 (Lonza) containing 36 ng/ml hydrocortisone (Sigma), 10 ng/ml EGF (Sigma) and 40 pg/ml tri-iodothyronine (Sigma) and 10% FBS. To mimic a diabetic environment and induce cellular insulin resistance in vitro, human cells were grown in the presence of 100 nmol/l insulin (Tocris, Bristol, UK), 25 mmol/l glucose (Sigma), 1 ng/ml TNF-α and 1 ng/ml IL-6 (R&D systems, Abingdon, UK)[17]. Cells were studied after 10–12 days of differentiation at 37 °C and were free of Mycoplasma infection. For insulin stimulation experiments cells were initially insulin and serum starved for 4 h and then treated with either 10 or 100 nM of insulin for 15 min. No commonly misidentified cell lines were used in this study. All kidney cell lines were generated by us as outlined in references included[20–23].

**Generation of stable cell lines.** IR-overexpressing cell lines were generated by transduction with human IR-containing lentivirus and blasticidin selection as previously described[17]. Briefly, lentiviral particles were made by subcloning human IR (NM_000208.2) into pLenti-TetCMV(IR)-Rsv(RFP-Bsd) vectors (Gentarget, San Diego, CA, USA), which were transfected into Lenti-X 293T cells (Clontech/Takara Bio Europe SAS, Saint-Germain-en-Laye, France), alongside pMD.2G (Addgene no. 12259) and psPAX2 (Addgene no. 12260), both gifts from D. Trono (École polytechnique fédérale de Lausanne). NRBF2-knockdown cell lines were also generated using lentiviral vectors expressing shRNA targeting human *NRBF2* (or scramble control) and a puromycin selection (designed and purchased with 'VectorBuilder').

### Glucose transport
Cellular glucose uptake was measured as previously described[17]. Briefly, cells were serum-starved before incubation with a modified Krebs Ringer Phosphate (KRP) solution for 15 min at 37 °C. After appropriate stimulation, [3H]2-deoxy-d-glucose (Perkin Elmer, Coventry, UK) was added at 37 kBq/ml for 5 min. Solubilised cell suspensions were collected, and radioactivity was measured in disintegrations per minute (dpm) using a multi-purpose scintillator counter (Beckman Coulter, High Wycombe, UK). Each condition was performed with at least two technical replicates. Data are presented as mean ± SEM. One-way ANOVA with Tukey's multiple comparison test and t-tests for statistical significance were performed using GraphPad Prism v9 (GraphPad Software, CA, USA).

### Western blotting
Briefly, total protein lysates were extracted using RIPA lysis buffer (Sigma Aldrich), resolved using SDS–PAGE and blotted onto PVDF membranes. Membranes were incubated in primary antibodies overnight at 4 °C, before washing and incubation with the appropriate horseradish peroxidase (HRP)-conjugated secondary antibody (Sigma Aldrich) at a 1:10,000 dilution. Primary antibodies were diluted 1:1000 in BSA and used to target total Akt, phospho-Akt (S473), phospho-IGF-I Receptor β (Tyr1135/1136)/insulin receptor β (Tyr1150/1151), total insulin receptor β and NDUFB8 (Cell Signalling Technologies). ATP5A, UQCRC2, SDHB and COXII were targeted using a total human OXPHOS antibody cocktail (Abcam). Immunoreactive bands were visualised using Clarity ECL Western Blotting Substrate (Bio-Rad, Hemel Hempstead, UK) on an AI600 imager (GE Healthcare, Amersham, UK) and quantified using ImageJ (NIH, https://imagej.nih.gov/ij/). Quantified western blotting data are presented as mean ± SEM. One-way ANOVA with Tukey's multiple comparison test and t-tests for statistical significance were performed using GraphPad Prism v9 (GraphPad Software, CA, USA).

### RNA extraction and real-time qRT-PCR
Total RNA was isolated using an RNeasy Mini Kit (QIAGEN, Germany) as per the manufacturer's recommendations. cDNA was synthesised using a high-capacity RNA-cDNA kit (Thermo Fisher Scientific, UK). Quantitative RT-PCR was performed using SYBR green (Sigma Aldrich) in a StepOnePlus system (ThermoFisher Scientific) for human *NRBF2* (forward: CAGACGAGCAGACCGTTTATT, reverse: TGCTGGGCTTTCA ATCTTTGTT) *ATP5A* (forward: AACCAGCATCACACACACAC, reverse: CACCAGGATAGGACGAGGAC), *UQCRC2* (forward: TTTTGTCTGCTT CCTGTGCC, reverse: TCGGCAGTGTGTCAAAAGTG), *SDHB* (forward: AGAAACTGGACGGGCTCTAC, reverse: TGTGGCAGCGGTATAGAGAG), *COXII* (forward: ACCGTCTGAACTATCCTGCC, reverse: AGATTAG TCCGCCGTAGTCG), *NDUFB8* (forward: TGCTTAGCCCCATTTCCTGA, reverse: AAGTAGGGGTGGAGAAGTGC); all normalised to *B-ACTIN* (forward: CACCATTGGCAATGAGCGGTTC, reverse: TAGGTCTTTGC GGATGTCCACGT).

### RNA sequencing
Libraries were prepared and sequenced at Abbvie Research Centre (Cambridge, MA, USA). cDNA libraries were prepared using the Clonetech SMART-Seq v4 Ultra Low Input RNA Kit for Sequencing. All samples had RIN > 8, measured using the Agilent TapeStation. Sequencing was performed on the Illumina HiSeq 4000 platform (Illumina Inc.) and produced paired-end 75 bp reads, with replicates appropriately partitioned into the two batches and containing a common reference sample. The reads were aligned to the GRCh38 human reference genome using the STAR v2.6.0.c aligner on default settings[69]. The quality of the RNA-Seq samples was verified with

FASTQC version 0.12.1[70]. Read counts were quantified with htseq-count 0.9.1[71].

Genes with more than 1 cpm (count per million) from at least one library were considered for the statistical analysis. The read count data were normalised with the TMM (trimmed mean of M-values) method from the edgeR R package (v3.22.5)[72] and transformed to log2 cpm with the voom method from the limma R package (v3.38.3)[73]. The principal component analysis was done in R (v3.5.1).

Different linear models were built on the transformed data on each independent cell type on the two conditions with five replicates, each of 'Basal' and 'Insulin Resistant' with the overexpression of the human insulin receptor (IR). Differential expression analysis was then performed on each model, with the contrast representing the difference between 'Insulin Resistant' and 'Basal' using limma. $p$-values produced from the differential analysis were adjusted for the four cell types using the Benjamini and Hochberg correction[74], and adjusted $p$-values (FDR) < 0.05 were considered significant unless otherwise stated. For the identification of cell-line-specific changes, we used data that were individually normalised and analysed for each cell line to capture information from genes that displayed cell-line-specific expression patterns. Nominal $p$-values were used to investigate the overlap between regulated genes. Cell-line-specific regulated genes were further filtered such that in addition to $p > 0.05$ in the other cell lines, Log2FC < 0.1 for upregulated and >−0.1 for downregulated transcripts/ proteins were employed.

## Tandem mass tag (TMT)-mass spectrometry (MS) processing and analysis

Total cell protein was extracted in RIPA lysis buffer (Thermo Fisher) and aliquots of each sample were digested with trypsin (2.5 µg per 100 µg protein; 37 °C, overnight), labelled with Tandem Mass Tag (TMT) ten plex reagents according to the manufacturer's protocol (Thermo Fisher Scientific, Loughborough, LE11 5RG, UK). Labelled samples were pooled and 50 µg was desalted using a SepPak cartridge (Waters, Milford, MA, USA). Eluate from the SepPak cartridge was evaporated to dryness and resuspended in 20 mM ammonium hydroxide, pH 10, prior to fractionation by high pH reversed-phase chromatography using an Ultimate 3000 liquid chromatography system (Thermo Fisher Scientific). The sample was loaded onto an XBridge BEH C18 Column (130 Å, 3.5 µm, 2.1 mm × 150 mm, Waters, UK) and peptides eluted with an increasing gradient (0–95%) of 20 mM ammonium hydroxide in acetonitrile, pH 10, over 60 min. The resulting fractions were evaporated to dryness and resuspended in 1% formic acid prior to analysis by nano-LC MSMS using an Orbitrap Fusion Lumos mass spectrometer (Thermo Scientific). High pH RP fractions were further fractionated using an Ultimate 3000 nano-LC system and spectra were acquired using an Orbitrap Fusion Lumos mass spectrometer controlled by Xcalibur 3.0 software (Thermo Scientific) and operated in data-dependent acquisition mode using an SPS-MS3 workflow. Replicates were appropriately partitioned into each batch with additional inclusion of a common reference sample, to allow batch correction. Raw data files for the total proteome analyses were processed and quantified using Proteome Discoverer software v2.1 (Thermo Scientific) and searched against the UniProt human database (September 2018: 152,927 entries) using the SEQUEST algorithm. The reverse database search option was enabled, and all data was filtered to satisfy a false discovery rate (FDR) of 5%.

The data output from the Proteome Discoverer 2.1 analysis was further handled, processed and analysed using Microsoft Office Excel, GraphPad Prism and R. Normalisation and differential analysis were performed in R in the same manner as RNA-seq data.

## Consensus OPLS

Consensus orthogonal partial least-squares discriminant analysis (OPLS-DA) model was computed with the MATLAB 9 environment with combinations of toolboxes and in-house functions that are available at https://gitlab.unige.ch/Julien.Boccard/consensusopls. Modified RV-coefficients were computed with the publicly available MATLAB m-file[75]. KOPLS-DA was assessed with routines implemented in the KOPLS open source package[76]. Consensus OPLS modelling was performed on proteomics and RNAseq data tables, which were all auto-scaled prior to the analysis for each cell type independently. The Consensus OPLS model distinguishes variations of data that are correlated to Y response (basal vs. insulin resistant) and those that are orthogonal to Y response. This eases the biological interpretation of results and enables the link between the variation of variables and the variation of the outcome while removing information coming from other sources of variation. The models were computed with two latent variables, 1 predictive and 1 orthogonal. The quality of the model was assessed by $R^2$ and $Q^2$ values, which define the portion of data variance explained by the model and the predictive ability of the model, respectively. The $Q^2$ value was computed by a $K$-fold cross-validation ($K = 7$). To ensure the validity of the model, a series of 1000 permutation tests were carried out by mixing randomly the original $Y$ response (basal vs insulin resistant). A $t$-test was performed to ensure that the true model $Q^2$ value was clearly distinguished and statistically different from the random model distribution. The variable relevance to discriminate between the two conditions was evaluated using the variable importance in projection (VIP) parameter, which reflects the importance of variables both with respect to the response and to the projection quality.

## Enrichment analysis

Over-representation analysis to identify shared transcription factors was performed using the TRANSFAC database v7.4 in WebGestalt 2019[77]. The Gene Ontology term enrichment analysis was performed with GSEA[26] using the R package clusterProfiler (v3.14.0)[78]. The GO annotation (biological process (BP), molecular function (MF) and cell compartment (CC)) for genes and proteins mapped to genes was obtained from the Molecular Signatures Database (MSigDB v7.1)[79]. The signal2noise ratio[26] was used for feature rankings in each omics from individual DE analysis in each cell type. The loadings for genes and proteins obtained from the Consensus OPLS multiblock model were used for feature rankings in each omics from the data integration in each cell type. The enriched GO terms were filtered for significance in at least one cell type (nominal $p$-value < 0.05 and $q$-value < 0.1 from both omics data either from individual DE or consensus OPLS), hierarchically clustered with the semantic similarity between GO terms based on the graph structure of GO (Wang measure) using the R package GOSemSim (v2.12.1)[27] and then displayed as a heatmap of normalised enrichment scores (NES) with the R package Complex-Heatmap (v2.2.0)[80]. Additional visualisation for consistently regulated pathways and respective core enrichment genes was performed in Cytoscape[81] using the EnrichmentMap module[82]. To assess whether similar pathway activation was evident in human DKD, we evaluated the expression of the core-enrichment genes for select clusters of regulated pathways (identified in insulin resistant cell lines), in the human kidney biopsy data outlined below. An average of the $Z$-scores was calculated for each of the core-enrichment genes to compare regulation in kidneys from early- or late-stage DKD with healthy living donors.

## Human cohorts

Insulin resistance-associated transcripts identified from cell-line studies were analysed in human kidneys using gene expression data from an American Indian type 2 diabetes cohort ($n = 69$ glomerular and $n = 47$ tubular samples)[28,83], and European Renal cDNA Bank–Kroener–Fresenius biopsy bank (ERCB, $n = 12$ glomerular and $n = 17$ tubular samples)[25,84] (Supplementary Data 3). We further used data from the 'Nephroseq' database (www.nephroseq.org, University

of Michigan, Ann Arbor, MI, USA) to correlate *NRBF2* expression with eGFR in late-stage DKD, using the 'Ju CKD' dataset[85]. Samples from the American Indian type 2 diabetes cohort consist of protocol human kidney biopsies from individuals with type 2 diabetes from the Gila River American Indian Community. The study participants were enroled in a randomised, double-blinded, placebo-controlled interventional clinical trial funded by the National Institute of Diabetes and Digestive and Kidney Diseases (NIDDK)[83]. All biopsies were stratified by the reference pathologist of the ERCB according to their histological diagnoses. Histology reports, clinical data, and gene expression information were stored in a de-identified manner. Both the American Indian type 2 diabetes cohort and ERCB biopsies were processed similarly; glomerular and tubular compartments were separated using microdissection and Affymetrix-based gene expression profiling was performed using Affymetrix GeneChip Human Genome U133A 2.0 and U133 Plus 2.0 Array (Affymetrix, Santa Clara, CA, USA), as previously reported[25,84]. Gene expression changes in DKD samples were compared to similarly processed healthy control biopsies using the significance analysis of microarrays (SAM) method implemented in the TIGR MultiExperiment Viewer application. Genes regulated with a *q*-value (false discovery rate) < 0.05 were considered significant. Spearman correlation was applied to evaluate the association between expression levels and phenotypes of interest.

### Analysis of single-cell RNA sequencing data
Cell-line-specific changes to insulin resistance were further evaluated in human single-cell RNA sequencing (scRNA-Seq) data from DKD, using data from 18 living kidney transplant donors (LD), 44 individuals with early DKD from an American Indian type 2 diabetes cohort[28] and 10 individuals with DKD downloaded from the Kidney Precision Medicine Project (KPMP, Supplementary Data 8)[86] tissue atlas (https://atlas.kpmp.org/repository). Details of tissue processing, single-cell isolation and scRNA-Seq are described in previous publications[86,87] and are according to the KPMP scRNA-Seq protocol (https://www.protocols.io/view/single-cell-rna-sequencing-scrna-seq-7dthi6n). In brief, kidney biopsies procured in CryoStor® were dissociated into a single-cell solution by enzymatic digestion for 12 min at 37 °C. Over 20,000 viable cells were run on the droplet-based 10× Genomics platform applying the Chromium Single Cell 3'chemistry (v3.1). After cDNA library preparation, sequencing was executed on an Illumina NovaSeq 6000 platform with more than 200 million reads (paired-end 2 × 151 bases) per sample. Barcode processing, and gene expression quantifications were performed with the 10X Cell Ranger v3 pipeline using the GRCh38 (hg38) reference genome. The cell ranger count matrix files are then processed using SoupX (v1.5.0) to remove the ambient mRNA contamination[88]. Only cells that passed the threshold of >500 and <5000 genes were used for further analysis using the Seurat R package. Seurat processing steps include normalisation, scaling, dimensionality reduction (principal component analysis and uniform manifold approximation and projection), harmony integration and unsupervised clustering. Cell-type expression of individual genes in DKD vs. LD was visualised using the dotplot tools implemented in Seurat and cellXgene.

### Measurement of cellular energetics (Seahorse XF)
A Seahorse XFe96 Analyser (Agilent Technologies) was used to assess metabolic activity through oxygen consumption rate (OCR) and extracellular acidification rate (ECAR). Cell densities and chemical working concentrations were optimised prior to the experiments. All cells were seeded in an XF 96-well microplate at densities of 30,000 cells/well for podocytes and mesangial cells, 25,000 cells/well for proximal tubular cells and 40,000 cells/well for glomerular endothelial cells, incubated at 33 °C overnight, followed by incubation at 37 °C for 10 days. The sensor cartridge was hydrated overnight at 37 °C and cells were equilibrated in assay buffer at 37 °C in a non-$CO_2$ incubator

for 1 h prior to the assay. Three baseline measurements were taken, followed by sequential injection of oligomycin (1.5 μM), FCCP (1 μM), and antimycin A plus rotenone (0.5 μM) and the calculation of OCR and ECAR using Agilent Wave software version 2.6.3. For data normalisation, cells were subsequently stained with Hoechst, imaged and counted using an INCell analyser[89], and BCA assays were performed.

### Reporting summary
Further information on research design is available in the Nature Portfolio Reporting Summary linked to this article.

## Data availability
The codes are available at https://github.com/sib-swiss/BEAt_DKD. The transcriptomic and proteomic datasets from insulin-sensitive and insulin-resistant cell lines are submitted and will be made publicly available to NCBI under the BioProject PRJNA905899. Additional data used in this manuscript are accessible at https://epdc.sib.swiss (European Platform for Diabetes and Complications) and https://atlas.kpmp.org/repository (Kidney Precision Medicine Project). All participants from the human cohort provided informed consent. Due to privacy protection concerns, individual-level genotype, and gene expression data from the early DKD study cannot be made publicly available. Other available source data are provided as a Source Data file with this paper. Source data are provided with this paper.

## Code availability
Code used in the analysis and presentation are available at https://github.com/sib-swiss/BEAt_DKD.

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

## Acknowledgements

This project has received funding from the Innovative Medicines Initiative 2 Joint Undertaking (JU) under grant agreement No. 115974. The JU receives support from the European Union's Horizon 2020 research and innovation programme and EFPIA and JDRF. Any dissemination of results reflects only the author's view; the JU is not responsible for any use that may be made of the information it contains. The KPMP is funded by the following grants from the NIDDK: U2C DK114886, UH3DK114861, UH3DK114866, UH3DK114870, UH3DK114908, UH3DK114915, UH3DK114926, UH3DK114907, UH3DK114920, UH3DK114923, UH3DK114933, and UH3DK114937. Other funding for this project includes the Swedish Heart-Lung Foundation (20190470), Swedish Research Council (EXODIAB, 2009-1039; 2018-02837), Swedish Foundation for Strategic Research (LUDC-IRC, 15-0067) to MFG. *The KPMP is funded by the following grants from*

*the NIDDK: U01DK133081, U01DK133091, U01DK133092, U01DK133093, U01DK133095, U01DK133097, U01DK114866, U01DK114908, U01DK133090, U01DK133113, U01DK133766, U01DK133768, U01DK114907, U01DK114920, U01DK114923, U01DK114933, U24DK114886, UH3DK114926, UH3DK114861, UH3DK114915, UH3DK114937.* The Medical Research Council (Senior Clinical Fellowship) funded RJMC-MR/K010492/1) together with project grant funding MR/W019582/1 and MR/T002263/1. This work is partially supported by George O'Brien Michigan Kidney Translational Core Center at the University of Michigan, funded by NIH/NIDDK grant 2P30-DK-081943. A.C.L. is supported by a Kidney Research UK Intermediate Fellowship (INT_002_20220705).

## Author contributions

A.C.L., V.B., J.A.H., A.F.B., R.J.P.P., M.C.S., M.C.W. and K.J.H. performed laboratory experiments; A.C.L., R.J.M.C., S.C.S. and M.F.G., designed the experiments with contributions from V.B., J.A.H., A.F.B., R.J.P.P. and K.J.H.; A.C.L, V.D.T.T., V.N., F.B., F.M. and W.J. designed the analysis with contributions from M.K., P.L., K.J.H., M.I., S.C.S., M.F.G. and R.J.M.C.; V.D.T.T., V.N., A.C.L., F.B., A.A., D.K., F.M., P.L., K.J.H., R.G.N., W.J., and M.K. analysed data; F.B., K.J.H., W.J., M.K., M.I., S.C.S., M.F.G. and R.J.M.C. oversaw the analysis and provided guidance on the design and results; A.C.L. prepared figures with contributions from V.D.T.T., V.N., J.A.H., D.K., F.M., W.J., H.C.L., R.G.N., R.J.M.C. and M.K.; R.M. and E.O. conducted expression analysis in American Indian type 2 diabetic cohort, ERCB and KPMP; with input from H.C.L. and R.G.N.; A.C.L. drafted and edited the manuscript with contributions from R.J.M.C.; Facilitated overall collaborations in the project—BEAt-DKD consortium. All authors reviewed, commented on, and approved the final version.

## Competing interests

The authors declare no competing interests.

## Additional information

## BEAt-DKD consortium

**Abigail C. Lay** [1,2], **Van Du T. Tran** [3], **Viji Nair**[4], **Frédéric Burdet** [3], **Florence Mehl** [3], **Dmytro Kryvokhyzha** [5], **Abrar Ahmad** [5], **Mark Ibberson** [3], **Wenjun Ju**[4,6], **Matthias Kretzler**[4,6], **Simon C. Satchell** [1], **Maria F. Gomez** [5] & **Richard J. M. Coward** [1]✉

