## [Transparent Peer Review file · Nature Communications]

Integrative transcriptome and proteome profiling of insulin-resistant kidney cell models and human biopsies reveals common and cell-type-specific mechanisms underpinning Diabetic Kidney Disease

Corresponding Author: Professor Richard Coward

Version 0:

Reviewer comments:

Reviewer #1

(Remarks to the Author)

This manuscript presents a study where the authors conducted bulk RNA-seq and proteomics analyses on four different kidney cell lines (in vitro) to investigate the similarities and differences in gene expression changes between RNA and protein in response to insulin resistance. To gain further insights, they compared these molecular alterations to those observed in human DKD kidneys by re-analyzing publicly available bulk and single cell RNA-seq datasets. The study successfully identified several disease-associated genes (CTSS, NRBF2, C3, CXCL1, TFPI2, and PFKFB3) and pathways (inflammatory response, ER stress, and glycoprotein metabolism) that exhibited consistent changes between insulin-resistant kidney cell lines and human DKD biopsies. Additionally, the authors observed a defective mitochondrial function in insulin-resistant podocytes, mesangial cells (MCs), and proximal tubular cells (PTC).

Overall, the authors generate new RNA-seq and Mass Spec datasets for four different kidney cells under basal and insulin-resistant conditions. A limitation of the study is that the majority of molecular changes were observed in in vitro cell lines. However, the authors mitigated this limitation by comparing their in vitro findings with publicly available DKD human kidney data. To further enhance findings, the authors should incorporate additional validation data and improve the data interpretation. Some specific concerns are outlined below:

1. Figure 1B: Total Akt expression was dramatically reduced under DM condition + IR transduction in Pod and MC. Using total Akt as a loading control for quantification may not be appropriate. Please add another house keep gene such as tubulin/ b-actin as protein loading control.
2. Figure 1C-F: IR transduction alone significantly alters glucose uptake when insulin is not supplemented. The loss of insulin-stimulated glucose uptake among the kidney cell types might be due to the baseline change of glucose uptake after IR transduction. The authors need to emphasize that point.
3. Supplemental Figure 3: RGN is not a well-known PT marker. The authors should change it to other putative PT markers such CUBN, LRP2 and SLC34A1.
4. Figure 2C-F: Volcano plot to show the differential genes may not be a good approach. For example there are so many volcano plots that gene font sizes are too small to read. It may be more informative if the authors can add Venn diagrams to show the proportion of genes that are concordant or discordant between RNA and protein measurements.
5. Figure 3D: The correlations between GFR and NRBF2 are positive in both glom (0.25) and tubule (0.28). This result means that lower GFR has lower NRBF2 expression. However, the NRBF2 expression is upregulated in both early- and late- DKD kidneys which presumably have lower GFR. The authors should provide explanation for the underlying factors that contribute to this inconsistent results.
6. Figure 4: From the perspective of targeted therapy, it would be more interesting if the authors can show the

pathways/metabolisms that are unique to glom or tubule. In addition, it would be necessary to compare these pathways with those identified in chronic kidney disease (CKD) by Qiu et al. (PMID: 30275566) using other genetic analysis approaches.

7. Figure 5: In the same cell type, the DE genes identified from RNA and protein profiling are quite different. Does this inconsistency stem from the discordance between transcription and translation processes or if it is due to technical artifacts? To address this issue, the authors can select one of these discordant genes for each cell type and validate them by qPCR (RNA) and western blot (protein).

8. Figure 6C-J: The authors should compare the expression of these selected genes between healthy and DKD. This analysis will help determine if the observed changes in gene expression in the in vitro cell culture model are consistent with the alterations observed in human DKD.

9. Figure 7: The authors should select specific mitochondrial genes that were identified from the RNA or protein analysis and perform validation experiments using qPCR and western. This validation is crucial to assess whether the expression changes of these mitochondrial genes are solely observed at the protein level without significant alterations at the RNA level.

10. The figures provided in the study appear to be of low resolution, which compromises the ability to accurately assess the quality of the data presented.

Reviewer #2

(Remarks to the Author)

In this manuscript, the authors take a shotgun profiling approach towards 4 kidney cell types of interest involved in diabetic kidney disease. They profiled the transcriptome and proteome of immortalized human podocytes, mesangial, glomerular endothelial, and proximal tubule cells. Bioinformatic analyses focused on insulin resistance DEGs and pathway enrichment or depletion shared on the transcriptomic and proteomic level. The authors point towards common enrichment of pathways that are largely known and not novel, as well as cell type-specific regulation of mitochondrial function on the protein but not transcript level, which is interesting. The authors try to validate their findings in microdissected human kidney of early and late stage DKD as well as single-cell datasets from DKD patients. Mitochondrial dysfunction is validated in seahorse experiments.

This is a nice and interesting but mostly descriptive study largely relying on an in vitro model analyzing immortalized cells artificially transferred into a state of insulin resistance. The datasets will be of interest to the DKD community and should be provided in a searchable web database. Given the comments below, I would argue the manuscript receives a major revision before publication.

I have the following concerns:

1. For the validation of cell type specificity in single-cell datasets, the authors only included cells from their 4 cell types of interest. They should, of course, show the expression levels in all other cell types of the kidney if they want to prove specificity. Otherwise, the results presentation might be highly misleading. No methodological detail was provided how the single-cell dataset was created/subset/QC'ed etc.
2. The authors only validate single genes in their human single cell dataset. Can they also demonstrate pathway enrichment/depletion in early/late DKD vs. controls? This might actually be more insightful and representative than looking at the single gene level.
3. Also, I suggest the authors include validation experiments in human DKD and healthy control tissue (which is available to the KPMP consortium). This could include, e.g., staining on the protein or mRNA level the top targets they have identified in their analysis and demonstrating cell type specificity with double stainings.
4. I was disappointed by the absence of any mechanistic studies leveraging the immortalized in vitro system available to the authors. They should go back and demonstrate with siRNA or pharmacological treatment some of the effects they claim to have found in their bioinformatics and seahorse analyses. For example, do the authors see a rescue when targeting TYCF21, RASL11B, MGP, TWF2, BDH2, S100A1, and PIGR?
5. Given the lack of mechanistic insight, I feel the in vivo/human validation should be much more comprehensive to convince the readers that the effects analyzed in immortalized cells cultured in a highly artificial manner on plastic actually represent in vivo changes in patients. The authors should clearly acknowledge these limitations in the discussion.
6. Suppl. Fig. 2a-b demonstrates that proteome sample distances are not as clean as that of the transcriptome among podocytes. Can the authors comment on or have they analyzed in more depth the seemingly present uniqueness of some of these cell states among podocytes?
7. I fail to recognize the clear separation of insulin-resistant cells that the authors claim in their analysis of PCAs within individual cell populations (Suppl. Fig. 2c-d).
8. I do not understand how the authors picked the 3 pathway "clusters" presented in Figure 4. Can they explain in more detail or add an informational cartoon to demonstrate why they picked these 3? How do the data presented in this graph advance the story?
9. The authors should give p values for their correlation analyses between RNA and protein. GO GSEA heatmaps lack color legends. Some plots lack sufficient legends (e.g., Suppl. Figs. 2c-d).
10. All codes used to produce the results and figures presented should be made available in a repository such as GitHub or

similar.

Reviewer #3

(Remarks to the Author)

The authors performed a beautiful integrative analysis generating the common and cell-type specific transcriptome and proteome changes in 4 different insulin resistant kidney cell line models, and further validate their findings in human DKD cohorts.

Major revision:

1. For the goal of open science and better replication, the authors should provide more statement on data and code availability

1) In the data availability part, the authors stated that the RNA-Seq and TMT proteomics data are stored in NCBI BioProject PRJNA905899. I searched the ID PRJNA905899 in NCBI BioProject website (<https://www.ncbi.nlm.nih.gov/bioproject>), but it told me "No items found". Will the authors tell me how to actually access those data.

2) I believe that the authors only stated the availability of the RNA-Seq and TMT proteomics data from cell lines. What about the data from human cohorts? The authors should tell us where to retrieve those data if available, and if not available, please provide the reason. I would also suggest a drawing a demographic summary table of the subjects in each cohort. Also, GEO IDs such as (GSMxxx) or the KPMP IDs for each sample include in this study should be provided.

3) The author should provide a code availability statement. Whether or not they would share the code to replicate the study.

2. The authors should provide a more detailed description of the human cohort samples included in this study, specifically the method part "Human cohorts". How many early DKD bulk transcriptome data were generated from PIMA cohort? How many advance DKD bulk transcriptome data were generated from ERCB cohort? Was the bulk transcriptome data of living donor (n=18 as pointed in Figure 4) from ERCB cohort or the PIMA cohort?

3. In figure 4, since LD, early DKD and advanced DKD bulk transcriptome data were from different cohorts (PIMA Indian and ERCB), how the authors processed the data to make them comparable across different cohorts? Did they directly merge the transcriptome data from PIMA and ERCB, and then calculated the z score? Or did they simply calculate the z score within each cohort? Using either method, I would be concerned about the credibility of the results comparing transcriptome data from different cohorts. The same concern also arise in Figure 3C. It seemed that early DKD (from PIMA Indian) and advanced DKD (from ERCB) were both compared to living donors (n=18). Was the living donor belong to PIMA or ERCB cohort, or from a third dataset?

4. For the cell line transcriptome and proteome experiments. There are totally 4 (cell type)×4(condition)×5(sample)=80 samples for each transcriptome and proteome experiments. Did those experiments done in one batch? If they were done in different batches, did each batch balanced the insulin resistance VS insulin sensitive group? What method the authors used to mitigate batch effects? For example, if the insuline resistance GEC were in one batch, and the insulin sensitive GEC were in another batch, it would be difficult to determine whether the transcriptome difference between two conditions were insulin resistant effect or batch effect.

5. From figure 7a, it is interesting that in Pod and PTC that mitochondrial related genes were up-regulated while protein level were down-regulated in response to insulin resistance. What's the potential mechanism mediating such discrepancy? Is it possible to visualize how those mitochondrial related genes were coordinated changed (using method such as z score) in human cohorts comparing LD, early DKD and advanced DKD in both bulk transcriptome and single cell transcriptome data?

6. Line 110-112, "NPHS2, PECAM, EBF1 and RGN were examples of cell-specific genes that were solely detected in Pods, GECs, MCs and PTCs, respectively (Supplementary Fig. 3)." The author should clarify the expression levels of the above proteins in different cell lines by Westernblot to further support the results of bioinformatics.

7. Line146-158, the author found that C3, CXCL1, CTSS, NRBF2, PFKFB3 and TFPI2 are closely related to the progression of DKD. However, the role of NRBF2 in DKD is currently unknown. We suggest that the authors should detect NRBF2 expression levels in immortalized cell lines or human DKD patient biopsy specimens by WB OR IHC.

Version 1:

Reviewer comments:

Reviewer #1

(Remarks to the Author)

The authors have addressed all my concerns.

Reviewer #2

(Remarks to the Author)

The authors are to be commended for their additional work on this MS. Some specific points that may warrant minor revisions:

1. The question of cell type specificity of some of their cell type "target" markers remains. If anything, Suppl. Fig. 15 and 16 now demonstrate that the markers from their cell lines hardly align with the single-cell data. For example, MGP, which the authors use as an EC marker, is shown to be highest expressed in interstitial cells. In addition, it is also expressed in PECs and podocytes, so specificity is an issue. Also, MGP was only weakly expressed in early DKD endothelial cells. TWF2 is actually highly expressed in immune cells. TCF21 is fine regarding podocyte specificity. I think the discussion on cell type specificity is not an academic one, because if down the line, targeting approaches are to take away information from this

work, cell type specificity is highly important. This should be clearly acknowledged as a limitation and discussed at more length.

2. New Figure 4 is a good start for validation and I agree the new data strengthen the manuscript. Right now, it still feels somewhat naked and might be supplemented with additional analyses (qPCR, WB) that would substantiate the authors' hypothesis that "podocytes were protected against actin cytoskeletal changes" upon NRBF2 overexpression. Overall, this will be a nice resource for the community.

Reviewer #3

(Remarks to the Author)

I have reviewed the revised manuscript titled "Integrative transcriptomic and proteomic profiling of human insulin-resistant kidney cell-lines and biopsies reveals novel mechanisms underpinning DKD". The authors have addressed my previous concerns and comments carefully and effectively. The revisions have significantly improved the quality and clarity of the paper. I am satisfied with the changes made and agree to accept the revised manuscript.

Version 2:

Reviewer comments:

Reviewer #2

(Remarks to the Author)

The authors have sufficiently addressed all comments. Congrats on the nice paper!

Reviewer #1 (Remarks to the Author):

This manuscript presents a study where the authors conducted bulk RNA-seq and proteomics analyses on four different kidney cell lines (in vitro) to investigate the similarities and differences in gene expression changes between RNA and protein in response to insulin resistance. To gain further insights, they compared these molecular alterations to those observed in human DKD kidneys by re-analyzing publicly available bulk and single cell RNA-seq datasets. The study successfully identified several disease-associated genes (CTSS, NRBF2, C3, CXCL1, TFPI2, and PFKFB3) and pathways (inflammatory response, ER stress, and glycoprotein metabolism) that exhibited consistent changes between insulin-resistant kidney cell lines and human DKD biopsies. Additionally, the authors observed a defective mitochondrial function in insulin-resistant podocytes, mesangial cells (MCs), and proximal tubular cells (PTC).

Overall, the authors generate new RNA-seq and Mass Spec datasets for four different kidney cells under basal and insulin-resistant conditions. A limitation of the study is that the majority of molecular changes were observed in in vitro cell lines. However, the authors mitigated this limitation by comparing their in vitro findings with publicly available DKD human kidney data. To further enhance findings, the authors should incorporate additional validation data and improve the data interpretation. Some specific concerns are outlined below:

Thank you for this excellent summary and very helpful suggestions. One point we apologise for not making clear initially is that not all the human data is freely publicly available but was obtained through collaboration with Matthias Kretzler's team at University of Michigan and Robert Nelson's team at NIDDK, Arizona.

1. Figure 1B: Total Akt expression was dramatically reduced under DM condition + IR transduction in Pod and MC. Using total Akt as a loading control for quantification may not be appropriate. Please add another house keep gene such as tubulin/ b-actin as protein loading control.

We thank the reviewer for this point, which should have been included in the original submission. Please see updated Figure 1b: we now include repeated and more representative western blots, with more equivalent total protein levels loaded per well, and β -actin as a loading control. We have also added additional blots to show insulin receptor phosphorylation (Tyr 1150/1151) to further illustrate the induction of insulin resistance in our cell-model culture conditions.

2. Figure 1C-F: IR transduction alone significantly alters glucose uptake when insulin is not supplemented. The loss of insulin-stimulated glucose uptake among the kidney cell types might be due to the baseline change of glucose uptake after IR transduction. The authors need to emphasize that point.

Thank you for the comment. We apologise for not being clear that all the cells in figure 1c-f had stable expression of the insulin receptor (IR). The difference shown in this figure is growing the cells in a "diabetic" environment (DM) which induces insulin resistance by virtue of media containing high glucose, high insulin and high pro-inflammatory cytokine levels (Lay *et al.* Diabetologia, 2017). You are correct that the basal level of glucose uptake appears to differ between "diabetic" and "non-diabetic" environments in the different cell types. We agree it is important to highlight this point.

We do also agree that changing IR levels may alter basal glucose uptake. To address this comment, we have added additional data (NEW Supplementary Figures 1m-1p) to demonstrate the percentage change in cellular glucose uptake between “wild-type” and “IR-transduced” cells in the absence of insulin stimulation (i.e., under basal conditions). This shows that in our cell models, IR transduction alone has no significant effect on glucose uptake under basal conditions. A comment has also been added to the text: please see lines 99-100. We also hope the addition of further blots showing phosphorylation of the insulin receptor (point 1 above) also helps to illustrate the development of insulin resistance in these cells.

3. Supplemental Figure 3: RGN is not a well-known PT marker. The authors should change it to other putative PT markers such CUBN, LRP2 and SLC34A1. Megalin.

Thank you for this comment and we apologise for not making it clear why we chose this marker. We chose to include *RGN* (Regucalcin), as this is an enzyme that is exclusively expressed in proximal tubular cells (PTCs) in the kidney and was specifically present only in our PTCs. Other data available to us which led us to include *RGN* as an important PTC-specific gene was the single nuclear RNA sequencing from <http://humphreyslab.com/SingleCell/> (Reviewer Figure 1a) and Human protein atlas (<https://www.proteinatlas.org/ENSG00000130988-RGN/tissue/kidney#img>, Reviewer figure 1b).

Reviewer Figure 1. *RGN* expression in human kidney.

a Data from Humphreys lab ‘Kidney Interactive Transcriptomics’ (Humphreyslab.com/SingleCell/) demonstrating *RGN* expression in human kidney single nuclear sequencing data (Wilson *et al.* 2019) and **b** data from ‘Human Protein Atlas’ (<https://www.proteinatlas.org/>) demonstrating high levels of *RGN* protein expression in proximal tubular cells.

Unfortunately, it appears that many apparently specific PT markers (*LRP2* (Megalin) and *MME* (CD10)) may be detected in multiple kidney cell types, which we found, and as illustrated below was also the case analysing the excellent freely available Humphreys’ lab kidney single nuclear data set (<http://humphreyslab.com/SingleCell/>.) (Reviewer Figure 2a-c). *SLC34A1* is a specific marker of PTC (Reviewer Figure 2d) but unfortunately, we did not find any mRNA or protein signal in any of our cell types. We absolutely acknowledge that human cell lines are not perfect models and can lose cellular markers over time. We have added this important point to our new section highlighting limitations of this study (lines 428-432).

Reviewer Figure 2. PT marker gene expression in human kidney single-nuclear sequencing

Human kidney single nuclear sequencing data (Wilson *et al.* 2019) from Humphreys lab ‘Kidney Interactive Transcriptomics’ (Humphreyslab.com/SingleCell/), demonstrating the expression of **a** *MME* (CD10) in proximal tubular cells and highly expressed in podocytes **b** *CUBN* (Cubilin) in all cell types, particularly proximal tubular cells, podocytes and parietal epithelial cells **c** *LRP2* (Megalin) in proximal tubular cells, podocytes and parietal epithelial cells and **d** *SLC34A1* specifically in proximal tubular cells.

4. Figure 2C-F: Volcano plot to show the differential genes may not be a good approach. For example there are so many volcano plots that gene font sizes are too small to read. It may be more informative if the authors can add Venn diagrams to show the proportion of genes that are concordant or discordant between RNA and protein measurements.

Thank you for this helpful suggestion. We have added additional Venn diagrams to updated Figure 2 and have updated the volcano plots to make them more readable (with fewer labels on each plot). We think that the volcano plots are still informative, to provide an overview of the level of differential expression fold changes and significance (FDR) in the different cell types.

5. Figure 3D: The correlations between GFR and NRBF2 are positive in both glom (0.25) and tubule (0.28). This result means that lower GFR has lower NRBF2 expression. However, the NRBF2 expression is upregulated in both early- and late- DKD kidneys which presumably have lower GFR. The authors should provide explanation for the underlying factors that contribute to this inconsistent results.

The reviewer is correct to point this out, thank you for prompting us to further explain this finding. In the initial submission, we showed that *NRBF2* was upregulated in both early and late-stage DKD (compared to healthy living donors) and there was a negative correlation between *NRBF2* and the GFR slope, meaning higher *NRBF2* expression was associated with steeper decline of renal function. However, we also demonstrated a positive correlation with GFR in early DKD (which may be seemingly inconsistent). We did not investigate the correlation between *NRBF2* and GFR in late-stage DKD at this time. We speculate that the modest positive correlation of *NRBF2* with measured GFR at time of biopsy in the early DKD cohort is due to the hyperfiltration status of these individuals with early DKD (please see NEW Supplementary Table 3 for baseline characteristics of the human cohorts-These early-stage patients had a mean GFR of 145 ml/min/1.73m² compared to 46 ml/min/1.73m² in late-stage patients).

We have now investigated *NRBF2* expression in additional studies of late-stage DKD in Chronic kidney disease (CKD) cohorts ('Ju CKD' data in the 'Nephroseq' database). This shows that both glomerular and tubular *NRBF2* expression has a negative correlation with GFR, (i.e., higher *NRBF2* expression is associated with lower GFR levels). We have updated the text to further clarify this result (updated lines 159-162) and include the associations between *NRBF2* expression and GFR in the late-stage DKD cohorts analysed in Ju *et al.* (New Supplementary Figure 7). We conclude that our data shows that higher levels of *NRBF2* are associated with lower GFR in late DKD and steeper slope of GFR decline.

Mechanistically, we found that low cellular levels of NRBF2 are detrimental in all 4 cell types studied (New Figure 4) - please see reviewer 2 (point 4) and reviewer 3 (point 7). We think this shows the importance of mechanistically investigating targets found in a multi-Omic approach as it appears the increased levels of NRBF2 in kidney cells are a protective response in diabetes.

6. Figure 4: From the perspective of targeted therapy, it would be more interesting if the authors can show the pathways/metabolisms that are unique to glom or tubule. In addition, it would be necessary to compare these pathways with those identified in chronic kidney disease (CKD) by Qiu *et al.* (PMID: 30275566) using other genetic analysis approaches.

Thank you for this suggestion. We agree it would be interesting to elucidate the pathways that differ between the glomerular and tubular compartments; however, we don't think this will be clearly therapeutically beneficial (as we do not know of methods of doing this). However, with

the developments in cell-targeted Adeno Associated Viral (AAV) gene therapy we think identifying cell-specific changes could ultimately have therapeutic benefit. Indeed, we have recently published promising mouse data showing the therapeutic benefit of targeting the podocyte with AAV gene therapy in nephrotic syndrome (PMID:37863058). This is one of the reasons that we chose to also study the cell-type specific responses (Updated Figure 6) and highlight the discordance in regulation of mitochondrial bioenergetic processes between kidney cell types (Updated Figures 7 and 8). Of note (and in relation to mitochondrial bioenergetic responses), we also see differences between the glomerular cell types (i.e., glomerular endothelial cells have a different response to podocytes and mesangial cells); which would not necessarily be captured by only looking at glomerulus vs tubular responses.

With regards to comparing our results with those identified using other approaches, we have added a comment on this in the discussion (please see lines 408-413) The noteworthy study by Qiu *et al.* (PMID: 30275566) generates glomerular and tubular eQTL datasets which were integrated with CKD GWAS. The authors found CKD-associated variant rs11959928 influenced tubular *DAB2* expression and that reduced *DAB2* was protective in animal models. While our study does not use information on human genetic variation (we instead focus on changes driven by an insulin-resistant environment), for interest, we include expression of *DAB2* in our dataset below (see Reviewer Figure 3), which indicates *DAB2* transcript and protein expression is reduced in insulin resistant proximal tubules. We would be happy to add this to the manuscript in supplementary data if deemed necessary.

Interestingly, the pathway enrichment analysis by Qiu *et al.* found tubular eGenes to be enriched for endo-lysosomal function and eGenes from both tubular and glomerular compartment to be enriched for 'autophagy' and 'mitochondrial degradation'. We have cited this study when discussing our findings on mitochondrial regulation (please see line 383-384).

Review Figure 3.

Expression of *DAB2* transcript and *DAB2* protein in insulin-sensitive and insulin-resistant human proximal tubular cells (PTCs) *in vitro*; $n=5$, $*p < 0.05$

7. Figure 5: In the same cell type, the DE genes identified from RNA and protein profiling are quite different. Does this inconsistency stems from the discordance between transcription and translation processes or if it is due to technical artifacts? To address this issue, the authors can select one of these discordant genes for each cell type and validate them by qPCR (RNA) and western blot (protein).

Thank you for raising this important point.

We have performed further validation (western blots and qPCRs) of some of the discordant genes, that appear to be downregulated at the protein level, yet remain unchanged at the RNA

level, specifically focusing on proteins important for mitochondrial function (please see updated figure 7 (g-i) and the detailed response to point 9 below). In updated figure 6, we also now demonstrate the DE genes (in response to insulin-resistance in a cell-type-specific manner) that are consistent at both RNA and protein level. Although, of course, our proteomics dataset does not contain information on all protein-coding genes that we detect in our transcriptomics data. Furthermore, we think it of interest that at the transcript level several long- non-coding transcripts are detected.

8. Figure 6C-J: The authors should compare the expression of these selected genes between healthy and DKD. This analysis will help determine if the observed changes in gene expression in the *in vitro* cell culture model are consistent with the alterations observed in human DKD.

We agree this is important and are sorry this was not clear in the initial submission. We think it may have partially been due to the poor resolution of Fig 6C-J. Please see updated figure 6 with improved resolution and updated supplementary figure 15. These figures show the gene changes we identified in our insulin resistant *in vitro* model that are consistently regulated in equivalent cell types in human DKD vs healthy controls, using scSEQ data.

9. Figure 7: The authors should select specific mitochondrial genes that were identified from the RNA or protein analysis and perform validation experiments using qPCR and western. This validation is crucial to assess whether the expression changes of these mitochondrial genes are solely observed at the protein level without significant alterations at the RNA level.

Thank you for this important suggestion. As mentioned above, we have now performed additional experiments to validate our discordant findings between regulation of mitochondrial bioenergetic genes at protein- vs RNA-level.

We performed western blotting to detect protein expression OXPHOS complex subunits (using an OXPHOS antibody cocktail and specific NDUFB8 antibody) and qPCR to detect expression of equivalent subunits at the RNA level in all 4 cell types. We studied complex I (NDUFB8) complex II (SDHB), complex III (UQCRC2) complex IV (COX II) and complex V (ATP5A). We found a reduction in OXPHOS complex at the protein level in podocytes (complex I and IV), mesangial cells (complex I and IV), and proximal tubular cells (complex IV) and, if anything, a trend towards increased expression of these OXPHOS subunits in GECs. We did not detect a significant reduction in any of the subunits at the RNA level, in any of the cell lines, consistent with our RNAseq data. We think this is broadly supportive of our transcriptomic and proteomic analysis.

We have now also updated Fig. 7 and Supplementary Fig. 17 to demonstrate the protein/RNA changes for all OXPHOS complexes, TCA cycle and glycolysis enzymes that we detected in our proteomics and transcriptomics dataset. We also now include a summary schematic.

10. The figures provided in the study appear to be of low resolution, which compromises the ability to accurately assess the quality of the data presented.

We apologise for this. We have updated all figures at higher resolution.

Reviewer #2 (Remarks to the Author):

In this manuscript, the authors take a shotgun profiling approach towards 4 kidney cell types of interest involved in diabetic kidney disease. They profiled the transcriptome and proteome of immortalized human podocytes, mesangial, glomerular endothelial, and proximal tubule cells. Bioinformatic analyses focused on insulin resistance DEGs and pathway enrichment or depletion shared on the transcriptomic and proteomic level. The authors point towards common enrichment of pathways that are largely known and not novel, as well as cell type-specific regulation of mitochondrial function on the protein but not transcript level, which is interesting. The authors try to validate their findings in microdissected human kidney of early and late stage DKD as well as single-cell datasets from DKD patients. Mitochondrial dysfunction is validated in seahorse experiments.

This is a nice and interesting but mostly descriptive study largely relying on an *in vitro* model analyzing immortalized cells artificially transferred into a state of insulin resistance. The datasets will be of interest to the DKD community and should be provided in a searchable web database.

Given the comments below, I would argue the manuscript receives a major revision before publication.

Thank you for your positive comments regarding our manuscript. We agree that our submission was descriptive and primarily a resource paper. However, in our updated manuscript, we have added further mechanistic insight by exploring the role of NRBF2 in podocytes, mesangial cells, glomerular endothelial cells and proximal tubular cells, which we hope increases its interest (please see (point 4 below) and reviewer 3 (point 7)

As described, the transcriptomic and proteomic datasets from insulin-sensitive and insulin-resistant cell lines are submitted to NCBI under the BioProject [PRJNA905899](https://www.ncbi.nlm.nih.gov/bioproject/PRJNA905899), and will be made available upon publication. Other data are also accessible at <https://epdc.sib.swiss> (European Platform for Diabetes and Complications) and <https://atlas.kpmp.org/repository> (Kidney Precision Medicine Project). All participants from human cohort provided informed consent. Due to privacy and data protection concerns, individual-level genotype and gene expression data from the early DKD (American Indian with type 2 diabetes) study cannot be made publicly available.

I have the following concerns:

1. For the validation of cell type specificity in single-cell datasets, the authors only included cells from their 4 cell types of interest. They should, of course, show the expression levels in all other cell types of the kidney if they want to prove specificity. Otherwise, the results presentation might be highly misleading. No methodological detail was provided how the single-cell dataset was created/subset/QC'ed etc.

Thank you for these important points and suggestions.

In our analysis of the single-cell data, we chose to focus on our 4 cell types of interest to show consistency between the changes we observe in our insulin-sensitive/insulin-resistant human kidney cells *in vitro* and changes observed in these cell types in human DKD. We have focused our analysis and paper on these 4 cell types, due to their established roles in early DKD pathogenesis.

That said, we agree that it is important to show the expression of these genes of interest in context of other kidney cells detected in single cell sequencing from advanced-stage DKD. We now include this information in **NEW supplementary figure 16**. For *TWF2* (example of a gene regulated in insulin-resistant MCs), the inclusion of immune cell subtypes in this analysis (which have high *TWF2* expression) reduce the observed differences between DKD and control in cell types of interest.

As we also detail above; it is important to again note that our *in vitro* dataset focuses on insulin resistance as an important molecular driver of DKD, and therefore may not capture or represent expression changes that are caused by other molecular drivers of this complex disease. We have added a comment on this to our “study limitations” section (**lines 432-434**).

We also acknowledge that extensive methodological detail of the single-cell dataset was not listed in our methods, we instead referenced previous publications and the published protocol from the Kidney Precision Medicine Project (<https://www.protocols.io/view/single-cell-rna-sequencing-scrna-seq-7dthi6n>). We now briefly cover these steps in our methods (**please see lines 617- 630**).

2. The authors only validate single genes in their human single cell dataset. Can they also demonstrate pathway enrichment/depletion in early/late DKD vs. controls? This might actually be more insightful and representative than looking at the single gene level.

This reviewer correctly points out that we used single-cell sequencing data to identify and present examples of single genes that are consistently regulated in response to insulin resistance *in vitro* and in human DKD in a cell-type specific manner. We also, importantly, demonstrate examples of three pathways that are enriched in glomeruli or tubulointerstitium from early- and/or late-stage DKD vs control tissue (**Figure 5**). These pathways were identified in our insulin-resistant kidney cell lines and are consistently enriched in human disease, using bulk transcriptomic data.

To address the question of whether pathways regulated in a cell-type-specific manner in insulin resistance were also regulated in human DKD, we have now performed an additional analysis, where we highlight KEGG pathways that are significantly regulated in a cell-type-specific manner in insulin resistance *in vitro* (**updated Supplementary Fig. 13c**) identified the “core enrichment genes” from those pathways and calculated Z-scores for expression in single-cell sequencing data from human DKD.

Using this approach, we found that the significant increase in GAP junctions was also evident in proximal tubular cells in DKD (**reviewer figure 4**). We have included these results as a proof-of-principle that we can detect consistent changes in pathway enrichment genes (identified in our *in vitro* models of insulin resistance) in equivalent cell types *in vivo*. However, given the current limitations of the single-cell-sequencing data (such as representation of rarer cell types and sequencing depth), we were unable to detect enough of the pathway-specific genes in a sufficient podocytes and mesangial cells to perform this analysis. In future studies, and with increasing availability of scSEQ data, a thorough investigation of pathway enrichment using single-cell multi-omics data will be required. It is currently beyond the scope of this study to perform this in-depth analysis, but we have added this important point to the discussion (please see **lines 408-413**).

Reviewer figure 4.

Box plot displaying mean Z-scores of expression for core 'Gap junction'-genes in proximal tubular cells from single-cell sequencing data from DKD vs Living donors

3. Also, I suggest the authors include validation experiments in human DKD and healthy control tissue (which is available to the KPMP consortium). This could include, e.g., staining on the protein or mRNA level the top targets they have identified in their analysis and demonstrating cell type specificity with double stainings.

Thank you for these suggestions. Our validation focuses on using transcriptomics data from human kidney tissue (healthy vs DKD), rather than staining at the protein or mRNA level in tissue sections which is, unfortunately, often only semi-quantitative. Additionally (and again, unfortunately) many antibodies are not optimised for staining tissue sections. We have considered using RNAscope to measure mRNA levels for targets of interest, but historic kidney tissue is often not prepared optimally for this technique. We have added a sentence on this in our updated limitations section (lines 434- 437).

4. I was disappointed by the absence of any mechanistic studies leveraging the immortalized in vitro system available to the authors. They should go back and demonstrate with siRNA or pharmacological treatment some of the effects they claim to have found in their bioinformatics and seahorse analyses. For example, do the authors see a rescue when targeting TYCF21, RASL11B, MGP, TWF2, BDH2, S100A1, and PIGR?

We apologise for not having more mechanistic studies in our initial submission. To rectify this and show the potential strength of our multi omics approach we have investigated the importance of *NRBF2* in podocytes (Pods), glomerular endothelial cells (GECs), mesangial cells (MCs) and proximal tubular cells (PTCs).

We chose to focus our efforts on *NRBF2*, as this gene had not previously been studied in DKD or any form of kidney disease, unlike the other targets that we found were changing across all insulin resistant cell types and in human DKD (*CTSS*, *C3*, *CXCL1*, *TFPI2* and *PFKFB3*). We generated *NRBF2* knock-down Pods, GEC, MCs and PTCs, using short-hairpin RNA and *NRBF2* overexpressing Pods using lentiviral constructs expressing human *NRBF2*. Our additional experiments are displayed in NEW Fig. 4 and supplementary Fig. 8. These results show that *NRBF2* knockdown (compared to control cells expressing 'scrambled' shRNA) results in marked morphological differences and evidence of vacuolisation in each of the 4 cell types, significant cell loss of GECs, MCs and PTCs and fewer but larger Pods, within 96 hours. We then focused on *NRBF2* over-expression in human podocytes and found this protected against morphological differences induced by a diabetic, insulin resistant environment.

Collectively, we think this example illustrates the potential power of this approach, using cell models to identify novel targets for focused validation in human cohorts, follow up mechanistic studies to ultimately identify potential therapeutic targets which could be exploited in the future

in a cell-type specific manner. We consider the addition of this data and greatly strengthens this project.

The other cell -specific genes highlighted above remain very valid and important molecular targets which will form the basis of multiple follow up studies, as mentioned in our article (lines 226-227 and 345-347).

5. Given the lack of mechanistic insight, I feel the in vivo/human validation should be much more comprehensive to convince the readers that the effects analyzed in immortalized cells cultured in a highly artificial manner on plastic actually represent in vivo changes in patients. The authors should clearly acknowledge these limitations in the discussion.

We have added discussion on the limitations of our study (please see lines 425-440 - 'Study Limitations') but also hope the reviewer agrees we now have included an exemplar of how this data may be progressed in the future; providing additional mechanistic insight into the role played by *NRBF2* in kidney cells.

6. Suppl. Fig. 2a-b demonstrates that proteome sample distances are not as clean as that of the transcriptome among podocytes. Can the authors comment on or have they analyzed in more depth the seemingly present uniqueness of some of these cell states among podocytes?

The sample distances in the proteomic data were indeed not as well distinguished as that of the transcriptomic data, among podocytes and proximal tubular cells. We had investigated different methods of normalization for proteomics and concluded on the TMM method and voom transformation. This yielded slightly better separation between these two cell lines, as applied to transcriptomics. Both 'omics' data showed an agreement on the closeness of the samples from the two cell lines, as shown in the PCA in (updated) Fig. 2a and New Supplementary Fig. 2a. These two cell lines were clearly separated in the third principal component in PCA, rather than the first two components, thus explaining the unclear grouping in the hierarchical clustering on sample distances. We added PCA on the third component in New Supplementary Fig. 2a.

7. I fail to recognize the clear separation of insulin-resistant cells that the authors claim in their analysis of PCAs within individual cell populations (Suppl. Fig. 2c-d).

We have now added rings to our PCA plots, to highlight the separation of the Insulin Receptor expressing, insulin-resistant cell lines with regards to the other samples. Please see updated Supplementary Figure 2d-e. We have removed the work "clear" from the description of the data (line 109)

8. I do not understand how the authors picked the 3 pathway "clusters" presented in Figure 4. Can they explain in more detail or add an informational cartoon to demonstrate why they picked these 3? How do the data presented in this graph advance the story?

Thank you for prompting further clarification. We have updated the text with more explanation of how the pathways were detected (please see lines 177-179)

The pathways displayed in figure 5 (previously figure 4) were consistently upregulated in all insulin-resistant cell lines at the RNA and protein level. These 'pathway clusters' were identified from our GOBP-GSEA results, which are presented in the heatmap in Supplementary figure 9. As detailed in the methods (please see lines 569-587) enriched GO terms were filtered for significance in at least one cell type (nominal p-value<0.05 and q-value<0.1 from both transcriptomics and proteomics, either from individual DE or Consensus OPLS). Terms were then hierarchically clustered with the semantic similarity between GO terms based on the graph structure of GO (Wang measure) using the R package GOSemSim (v2.12.1) and then displayed as a heatmap of normalised enrichment scores (NES)).

Other examples of individual GO terms consistently regulated between cell types are highlighted in new Supplementary figure 12 and lines 195-197 (i.e., those where hierarchical clustering did not identify more than 2 similar GO terms with consistent regulation between insulin resistant cell types). These include 'Hippo signalling', 'regulation of pinocytosis', 'Signal transduction in response to ammonium ion', and 'iron ion transport'.

The data presented in the graphs for Fig. 5 (parts b, c, e, f, h, i) demonstrate Z-scores of expression for the 'core enrichment genes' for each pathway, using glomerular and tubular transcriptomics data from early- and late-stage DKD samples. Overall, our figure re-enforces potential importance of these pathways in DKD pathogenesis and that we observe consistent regulation between our cell lines and human disease for these responses.

9. The authors should give p values for their correlation analyses between RNA and protein. GO GSEA heatmaps lack color legends. Some plots lack sufficient legends (**e.g., Suppl. Figs. 2c-d**).

Thank you for pointing this out. We apologise for omitting some of this important information in our initial submission. We have updated Supplementary Figure 2 and the legend has been updated. We now provide further details in all of the figure legends and added p values for the correlation analysis (Fig 3a). The colour legends for the GO-GSEA heatmaps are included in the PDF (Supplementary Figures 9-11) and indicate the normalised enrichment score (NES).

10. All codes used to produce the results and figures presented should be made available in a repository such as GitHub or similar.

We agree and have made our codes available at https://github.com/sib-swiss/BEAt_DKD.

Reviewer #3 (Remarks to the Author):

The authors performed a beautiful integrative analysis generating the common and cell-type specific transcriptome and proteome changes in 4 different insulin resistant kidney cell line models, and further validate their findings in human DKD cohorts.

We thank this reviewer for their positive comments on our manuscript and helpful suggestions, which we respond to below.

Major revision:

1. For the goal of open science and better replication, the authors should provide more

statement on data and code availability. In the data availability part, the authors stated that the RNA-Seq and TMT proteomics data are stored in NCBI BioProject PRJNA905899. I searched the ID PRJNA905899 in NCBI BioProject website (<https://www.ncbi.nlm.nih.gov/bioproject> [[ncbi.nlm.nih.gov](https://www.ncbi.nlm.nih.gov)]), but it told me “No items found”. Will the authors tell me how to actually access those data.

We have updated our statement on data and code availability (please see lines 657 - 664)

2) I believe that the authors only stated the availability of the RNA-Seq and TMT proteomics data from cell lines. What about the data from human cohorts? The authors should tell us where to retrieve those data if available, and if not available, please provide the reason. I would also suggest a drawing a demographic summary table of the subjects in each cohort. Also, GEO IDs such as (GSMxxx) or the KPMP IDs for each sample include in this study should be provided.

As described above, the transcriptomic and proteomic datasets from insulin-sensitive and insulin-resistant cell lines are submitted and available to NCBI under the BioProject PRJNA905899. Other data are also accessible at <https://epdc.sib.swiss> (European Platform for Diabetes and Complications) and <https://atlas.kpmp.org/repository> (Kidney Precision Medicine Project). All participants from human cohort provided informed consent. Due to privacy protection concerns, individual-level genotype and gene expression data from the early DKD study cannot be made publicly available.

We have taken your very useful suggestion to present a demographic summary table of each cohort and now include these as Supplementary Tables 3 which are referred to in our updated methods (line 592).

We also include the KPMP IDs for the samples that we include in our study (please see Supplementary Table 8).

3) The author should provide a code availability statement. Whether or not they would share the code to replicate the study.

We agree. Please see statement above, all code is, or will be made, available at https://github.com/sib-swiss/BEAt_DKD.

4. The authors should provide a more detailed description of the human cohort samples included in this study, specifically the method part “Human cohorts”. How many early DKD bulk transcriptome data were generated from PIMA cohort? How many advance DKD bulk transcriptome data were generated from ERCB cohort? Was the bulk transcriptome data of living donor (n=18 as pointed in Figure 4) from ERCB cohort or the PIMA cohort?

We have now added additional details to the manuscript, including a demographics table for these cohorts (please see new Supplementary table 3). We have updated our methods section and, for each analysis, details of the sample numbers can be found in respective figure legends. In this study, we analysed bulk transcriptome data from $n=69$ (glomeruli) and $n=47$ (tubules) for early DKD biopsies (American Indian cohort) and $n=12$ for glomerular and $n=17$ for tubular transcriptomic data from advanced DKD cohorts. The living donors used in this study is not part of the Native American or ERCB cohort. These are living donor transplant biopsies obtained at the time of transplantation used as controls. (Yasuda *et al.* 2006 *Clin. Exp. Nephrol.* PMID:16791393) Biospecimens from all cohorts were collected after informed consent and with the approval of local ethics committee from all cohorts.

5. In figure 4, since LD, early DKD and advanced DKD bulk transcriptome data were from different cohorts (PIMA Indian and ERCB), how the authors processed the data to make them comparable across different cohorts? Did they directly merge the transcriptome data from PIMA and ERCB, and then calculated the z score? Or did they simply calculate the z score within each cohort? Using either method, I would be concerned about the credibility of the results comparing transcriptome data from different cohorts. The same concern also arise in Figure 3C. It seemed that early DKD (from PIMA Indian) and advanced DKD (from ERCB) were both compared to living donors (n=18). Was the living donor belong to PIMA or ERCB cohort, or from a third dataset?

The reviewer has rightfully pointed out an important issue and is indeed correct about combining different datasets. We share similar concerns and therefore have tried to address this in our pipeline. A large source of batch-to-batch variation are differences in sample processing (Leek et al, 2010 *Nature Rev Genet* PMID:20838408). To address this and minimize the batch effects, samples were collected and processed centrally following a harmonized protocol including consistent reagents, technologies, and even personnel where feasible. For the living donor samples, although from another cohort, we had harmonized the sample collection and sample preparation protocol.

To account for combining the samples, we also have always included “bridge samples” in each of our batches, which help when applying batch correction methods and to allow for effective merging of the datasets, inspection for batch effects, and addressing those with established tools (Zhang et al, 2018, *BMC Bioinformatics* PMID:30001694).

All these measures should reduce the variation due to technical differences/artifacts while maintaining the true biological signal, and allow the comparison of cohorts, although no method so far can eliminate all these biases. We have also added a discussion of these limitations (please see ‘Study limitations’ line 437- 440)

6. For the cell line transcriptome and proteome experiments. There are totally 4 (cell type)×4(condition)×5(sample)=80 samples for each transcriptome and proteome experiments. Did those experiments done in one batch? If they were done in different batches, did each batch balanced the insulin resistance VS insulin sensitive group? What method the authors used to mitigate batch effects? For example, if the insulin resistance GEC were in one batch, and the insulin sensitive GEC were in another batch, it would be difficult to determine whether the transcriptome difference between two conditions were insulin resistant effect or batch effect.

This is another important point which we addressed in our experimental design. The transcriptome samples were performed in one batch. The proteome samples were in two batches for each cell type, with replicates appropriately partitioned into the two batches and containing a common reference sample, which allowed for batch correction performed in the limma model construction. Please see Supp. Fig. 2e. We have also added a comment on this to our methods section (please see lines 539-540).

7. From figure 7a, it is interesting that in Pod and PTC that mitochondrial related genes were up-regulated while protein level were down-regulated in response to insulin resistance. What’s the potential mechanism mediating such discrepancy? Is it possible to visualize how those mitochondrial related genes were coordinated changed (using method such as z

score) in human cohorts comparing LD, early DKD and advanced DKD in both bulk transcriptome and single cell transcriptome data?

We agree that this interesting finding required further attention and explanation, and (as we note above for reviewer 1 point 9) we performed additional validation of this finding and present an updated Fig. 7 and Supplementary Fig. 17b and c.

There are several potential mechanisms which may mediate the discrepancy between RNA and protein abundance of the mitochondrial genes, for example increased protein degradation or impaired translation. As many of these protein complexes reside within the mitochondria, this may also reflect a reduction in functional mitochondria (which may occur via a process of mitophagy, for example). We discuss potential biological mechanisms for our finding within the discussion, please see lines 377-378 and 402-403

Unfortunately, we do not think (optimal) further validation of our mitochondrial findings in human cohorts is practical at present as this would require transcript and protein information from individual cells. Spatial proteomics will not have the required resolution to distinguish between individual glomerular cell types.

8. Line 110-112, "NPHS2, PECAM, EBF1 and RGN were examples of cell-specific genes that were solely detected in Pods, GECs, MCs and PTCs, respectively (Supplementary Fig. 3)." The author should clarify the expression levels of the above proteins in different cell lines by Western blot to further support the results of bioinformatics.

We have now added western blotting to Supplementary Fig. 1 to show expression of protein markers in appropriate cell lines. We also, importantly, now include a "study limitations" section, where we acknowledge the problems faced when using cell lines (Lines 428-432).

7. Line 146-158, the author found that C3, CXCL1, CTSS, NRBF2, PFKFB3 and TFPI2 are closely related to the progression of DKD. However, the role of NRBF2 in DKD is currently unknown. We suggest that the authors should detect NRBF2 expression levels in immortalized cell lines or human DKD patient biopsy specimens by WB OR IHC.

Thank you for this suggestion. As this reviewer points out, the role of NRBF2 in the kidney had not been explored and its role in DKD unknown. We have looked at this in NEW Figure 4 and please see our detailed response to reviewer 2 point 4.

REVIEWER COMMENTS

Reviewer #1 (Remarks to the Author):

The authors have addressed all my concerns.

Thank you

Reviewer #2 (Remarks to the Author):

The authors are to be commended for their additional work on this MS. Some specific points that may warrant minor revisions:

Thank you

1. The question of cell type specificity of some of their cell type “target” markers remains. If anything, Suppl. Fig. 15 and 16 now demonstrate that the markers from their cell lines hardly align with the single-cell data. For example, MGP, which the authors use as an EC marker, is shown to be highest expressed in interstitial cells. In addition, it is also expressed in PECs and podocytes, so specificity is an issue. Also, MGP was only weakly expressed in early DKD endothelial cells. TWF2 is actually highly expressed in immune cells. TCF21 is fine regarding podocyte specificity. I think the discussion on cell type specificity is not an academic one, because if down the line, targeting approaches are to take away information from this work, cell type specificity is highly important. This should be clearly acknowledged as a limitation and discussed at more length.

We fully agree with this reviewer that Supp fig 15 and 16 show that the genes we focus on for their cell-type-specific regulation have high expression in other cell types, beyond the 4 cell lines we focus on in our manuscript. The intention of this section is to highlight the genes which we found to have cell-type-specific regulation, not necessarily cell-type-specific expression. We apologise this was not made clear and have amended the text **lines 203 and 216-217** to highlight that Supp Fig 14 looks initially at the subset of cells labelled as Pods, ECs, MCs and PTCs. We have removed the UMAPs for *MGP* and *TWF2* in this figure, as we agree that they are not good examples of cell-type-specific expression, even amongst these 4 cell types.

The discovery that there is cell-type regulation is potentially helpful therapeutically as the genes/pathways could be specifically targeted in a cell specific manner with gene therapy (Adeno Associated Virus [AAV] or other approaches) in the future.

However, with regards to *MGP* expression, our data show that this gene is selectively regulated in ECs in insulin resistant conditions and that this is consistently shown *in vivo* from our single cell sequencing analysis. We found that *MGP* expression is lower in ECs from DKD vs healthy living donors. Although interstitial cells show high *MGP* expression, we did not see a significant regulation of *MGP* in this cell type. Likewise, although our analysis demonstrates a high expression of *TWF2* in the immune cell subcluster, we

found that this gene was selectively regulated in MCs in DKD (Supp Fig. 16b and Fig 6f and j).

We have elaborated on this in the discussion of the results (**lines 350-351**) and in our study limitations (**lines 441-445**).

2. New Figure 4 is a good start for validation and I agree the new data strengthen the manuscript. Right now, it still feels somewhat naked and might be supplemented with additional analyses (qPCR, WB) that would substantiate the authors' hypothesis that "podocytes were protected against actin cytoskeletal changes" upon NRBF2 overexpression.

Thank you for this helpful comment. The purpose of this figure was "proof-of concept" that our non-biased poly OMIC approach could identify novel cellular targets in the evolution of Diabetic Kidney Disease (potentially therapeutic and as biomarkers). We think NRBF2 illustrates this as all knock-out cell lines showed significant cell death in "normal" culture conditions (compared with scramble control). We then examined podocyte over expression and again get a readout of protection of the cytoskeletal structure in a diabetic environment. In podocytes the integrity of the actin cytoskeleton is critical for function¹

We absolutely agree that we have not thoroughly examined the mechanisms underpinning this. We intend to do this going forward and are currently putting a project grant together to do this. We intend examining all four cell types (podocytes, mesangial cells, Glomerular endothelial cells and mesangial cells) for survival, motility, autophagic flux, mitochondrial function and ER stress in diabetic and non-diabetic settings. We then intend exploring mechanism by performing total and phosphor-specific proteomics on all of these samples. We will examine all 4 kidney cell types manipulated in 3 ways [scramble – NRBF2 knock-down – NRBF2 over-expression] in diabetic and non-diabetic culture conditions. We will then validate pathways or targets identified. We think examining this interesting finding comprehensively will be of great interest and is an independent paper due to the volume of work required. If deemed by the reviewer that extra information is required we are happy to get the cells out and do some blotting for actin related proteins etc but do not think will greatly add to this body of work. We have alluded to this important issue in the discussion (**lines 313-314**)

1. Blaine, J., and Dylewski, J. (2020). Regulation of the Actin Cytoskeleton in Podocytes. *Cells* 9. 10.3390/cells9071700.

Reviewer # 3(Remarks to the Author):

I have reviewed the revised manuscript titled “Integrative transcriptomic and proteomic profiling of human insulin-resistant kidney cell-lines and biopsies reveals novel mechanisms underpinning DKD” . The authors have addressed my previous concerns and comments carefully and effectively. The revisions have significantly improved the quality and clarity of the paper. I am satisfied with the changes made and agree to accept the revised manuscript.

Thank you

We have no reviewers comments to address as all 3 very happy with paper now.